# Dietary restriction modulates ultradian rhythms and autocorrelation properties in mice behavior

Jackelyn Melissa Kembro [1,2], Ana Georgina Flesia[3,4], Victoria América Acosta-Rodríguez [5], Joseph S. Takahashi [5,6] & Paula Sofía Nieto [3,7] ✉

Animal behavior emerges from integration of many processes with different spatial and temporal scales. Dynamical behavioral patterns, including daily and ultradian rhythms and the dynamical microstructure of behavior (i.e., autocorrelations properties), can be differentially affected by external cues. Identifying these patterns is important for understanding how organisms adapt to their environment, yet unbiased methods to quantify dynamical changes over multiple temporal scales are lacking. Herein, we combine a wavelet approach with Detrended Fluctuation Analysis to identify behavioral patterns and evaluate changes over 42-days in mice subjected to different dietary restriction paradigms. We show that feeding restriction alters dynamical patterns: not only are daily rhythms modulated but also the presence, phase and/or strength of ~12h-rhythms, as well as the nature of autocorrelation properties of feed-intake and wheel running behaviors. These results highlight the underlying complexity of behavioral architecture and offer insights into the multi-scale impact of feeding habits on physiology.

Animal behaviors can be conceptualized as part of a complex system. Such systems found widely in nature from neuroscience to economics, share simple defining features: a large number of elements that interact with each other through non-linear relationships. These systems involve events and information flowing across a wide range of temporal and spatial scales. The multiple levels of organization within a complex system can mutually affect each other, giving rise to emergent global patterns (spatial and/ or temporal). Additionally, these systems are sensitive to different environmental cues, displaying remarkably specific responsiveness[1,2].

In living organisms, complexity is a consequence of evolutionary self-organization, and their statistical properties can be captured by universal physical laws[3–10]. In this context, oscillatory theory provides a framework for characterizing dynamics of biological rhythms across a wide range of organisms, including bacteria and humans, which exhibit diverse periodicities[11]. Circadian rhythms are ~24 h oscillations in biological processes found from metabolism and physiology to genetics, generated by internal timekeeping mechanisms[12]. Such endogenous rhythms have evolved in response to predictable environmental changes, such as ambient lighting, temperature, and nutrient availability. Through the temporal organization of metabolism, physiology, and behavior, these timekeeping mechanisms enable organisms to synchronize their internal processes with environmental timing cues, facilitating optimal adaptation[12]. In mammals, the circadian system governs ~24h rhythms in behavior and physiology[13], and co-exists with other biological rhythms with shorter periods such as ultradian rhythms[14–17]. Evolutionarily distant species exhibit a strikingly similar pattern in the dynamic organization of rhythms in daily and ultra-dian domains[16]. For example, feeding and locomotor behaviors in mammals and birds can exhibit prominent daily and other biological rhythms (i.e.,

[1]Universidad Nacional de Córdoba (UNC), Facultad de Ciencias Exactas, Físicas y Naturales, Instituto de Ciencia y Tecnología de los Alimentos (ICTA) and Departamento de Química, Cátedra de Química Biológica, Córdoba, Córdoba X5000HUA, Argentina. [2]Instituto de Investigaciones Biológicas y Tecnológicas, Consejo Nacional de Investigaciones Científicas y Técnicas (CONICET)- UNC, Córdoba, Córdoba X5000HUA, Argentina. [3]Universidad Nacional de Córdoba, Facultad de Matemática, Astronomía, Física y Computación, Córdoba, Córdoba X5000HUA, Argentina. [4]Consejo Nacional de Investigaciones Científicas y Técnicas (CONICET), Centro de Investigaciones y Estudios de Matemática (CIEM, CONICET-UNC), Córdoba, Córdoba X5000HUA, Argentina. [5]Department of Neuroscience, Peter O'Donnell Jr. Brain Institute, University of Texas Southwestern Medical Center, Dallas, TX 75390-9111, USA. [6]Howard Hughes Medical Institute, University of Texas Southwestern Medical Center, Dallas, TX 75390-9111, USA. [7]Consejo Nacional de Investigaciones Científicas y Técnicas (CONICET), Instituto de Física Enrique Gaviola (IFEG, CONICET-UNC), Universidad Nacional de Córdoba, Córdoba, Córdoba X5000HUA, Argentina. ✉e-mail: paula.nieto@unc.edu.ar

ultradian)[3,5,17]. However, the origin and functional role of ultradian rhythms in mammals are not fully understood.

Animal behavior, when analyzed on timescales ranging from mere fractions of a second to hours is not random[16]. Rather, behavioral dynamics exhibit a high degree of (auto) correlation between closely spaced time points[3,5,8,10,18–21]. The strength of autocorrelation diminishes in a power-law fashion as the time-lag between data points increases. These long-range autocorrelations act as a distinctive fingerprint of long-term memory within the behavioral patterns of healthy individuals. The importance of dynamics is evident since factors such as aging, disease, or stress, potentially lead to attenuation or even loss of long-range (auto) correlation.

Here, we analyzed temporal dynamics of mammalian behavior as a complex system, by combining an advanced integrative five-step wavelet method, GaMoSEC, for rhythm detection and characterization[16] with detrended fluctuation analysis (DFA) to assess autocorrelation properties. This approach provides a framework for associating dynamical patterns (i.e. daily and ultradian rhythms and autocorrelations) occurring at very diverse spatial and temporal scales. Specifically, we focus on the multi-scale modulatory effects on behavioral dynamics of two types of dietary restriction Caloric and Temporal Restriction, (CR or TR, respectively). Prior studies have shown that these feeding patterns differentially impact daily behavioral rhythms, as well as metabolism over a 42-day period[22]. However, it remained unexplored whether other dynamical patterns, present at shorter temporal scales (i.e., ultradian rhythms and long-range correlations), are affected by these feeding schedules. Our hypothesis is that dietary restriction schedules impact the diversity of dynamical patterns found in feeding and wheel-running activities, beyond circadian behavior itself. Understanding how different feeding patterns integrally affect mammalian behaviors is fundamental because dietary restriction is the most robust and least invasive intervention known to promote healthier longevity[23,24].

We show that the presence of 12h-rhythms is highly dependent on the behavior (food-intake or wheel running), with marked individual variability. In addition, temporal restriction increases, while caloric restriction decreases, the presence of 12h- rhythms in the food-intake time series. For wheel-running activity, the 12h-rhythms are less sensitive to the feeding restriction protocol imposed. When present, dietary conditions can also dynamically modulate phase and/or strength of 12h- rhythms in both behaviors. Lastly, we show that long-range correlations were observed in the wheel running, but not in the food-intake, time series for scales below 100 min. These correlation properties are also susceptible to caloric and time-restricted feeding modulation. Overall, our results highlight an integral view of mice behavior, considering simultaneously the diversity of dynamical patterns and their responses to external perturbations, contributing to building up the behavioral architecture that composes them as complex systems.

## Results
### Characterization of behavior complexity (24 h and 12 h rhythms and autocorrelation properties) under ad libitum (AL)
We start by applying the GaMoSEC (Supplementary Note 1) to the first week of the food-intake and wheel running time series[22] (Fig. 1a, b) in which mice ($n = 30$) were maintained under a 12 L:12D cycle and ad libitum conditions. Representative actograms are presented in Supplementary Fig. 8. Characterization of presence of different behavioral patterns is summarized in Fig. 1 and Table 1. As expected, we detect entrained 24h-rhythms in both feeding and wheel-running activity in all animals studied (Fig. 1c–h, Table 1) with the acrophase falling within the dark period, and wheel running leading the phase by less than an hour (~42 min) (Table 1). There is a high correlation between all pairs of animals (Table 1, see example Supplementary Fig. 5) within each behavior studied, consistent with entrainment of 24h-rhythms to the L/D cycle.

Ultradian rhythms, with periods near 12, 8 and 6 h (Fig. 1c–h, Table 1) were detected depending on the behavior and the individual (see criteria for UR period detection in Supplementary Note 2, Supplementary Figs. 2–4). Ultradian rhythmicity is less prevalent than 24h-rhythms and are more

frequently found for wheel running than food-intake activity. Contrary to 24h-rhythms, when present, 12h-rhythms in food intake leads the phase by 1 h when compared to wheel running activity. Although positive correlation between animals is also observed for the ultradian rhythms when present (Table 1), the lower correlation values arise from increased variability between mice compared with the 24h-rhythms, suggesting less influence of the LD cycles on ultradian rhythmicity.

Bifurcation patterns are observed in the real part of the Morlet cwt plots (Fig. 1e, f and Supplementary Fig. 1c in Supplementary Note 1) obtained for both wheel running and food-intake time series. To understand these bifurcation patterns further together with the autocorrelation structure of the time series, we used DFA. DFA is a method for characterizing the scaling behavior (i.e., the type of autocorrelation properties) present in a time series. An overview of the DFA algorithm is presented in Supplementary Note 4, as well as an illustrative example of DFA analyses of two artificially constructed activity time series (Supplementary Fig. 12). Specifically, an example of a artificial time series with long-range correlations and its analysis with DFA is shown in Supplementary Fig. 12a, c, e. In counter position, the analysis of a time series with random noise (non-correlated) with an amplitude that varies periodically according to a circadian dynamics is presented in Supplementary Fig. 12b, d, f. These two artificially constructed activity time series can be a useful guide to help interpret the DFA results obtained from experimental data analyses shown below.

The fluctuation function estimated with DFA of representative experimental food-intake and wheel running time series are presented in Fig. 1 panels i, j (open black circles). Two distinct linear scaling regions were observed: one for time scales between 10 and 100 min (short time scales, region 1); the other for time scales between 5 and 36 h (long time scales, region 2). The slope estimated in these two linear regions represents the autosimilarity parameters, $\alpha_1$ and $\alpha_2$ (or short and long temporal scales, respectively; see an example of selecting the optimal region for α-estimation in Supplementary Fig. 13).

The food-intake activity presents a $\alpha_1$ of ~0.5 (Table 1) indicating random fluctuation or short-range correlations in the feeding pattern; whereas the $\alpha_2$-values either remains similar or decrease (Supplementary Note 4, Table 1), indicating anti-(auto) correlation (i.e., large activity values are more likely to be followed by low activity values, vice versa for low activity values). In contrast, the $\alpha_1$ of wheel running showed strong positive correlations, indicating that large activity counts are more likely to be followed by large activity counts (and vice versa), with even larger $\alpha_2$-values (Table 1 and Supplementary Note 3). The two scaling regions observed in DFA, can be associated with the probability distributions of the durations of behavioral events as well as the duration of inter-event periods (Light-colored filled histograms in Fig. 1i, j). Note, that the majority of the food-intake and wheel running events and inter-events presented durations predominantly lasting less than 100 min (Fig. 1i, j).

In sum, beyond the 24h-rhythms, mice under 12 L:12D cycle and AL conditions present a diversity of dynamical patterns in behavior (ultradian rhythms and short/long-range correlations), which integrally build up the behavioral architecture that compose these complex systems. These dynamical patterns present clear differences when analyzing running or feeding activity, which may reflect distinct underlying biological processes integrally involved in each behavior and also different interaction with environmental cues such as the LD cycle.

### Modulation of 24h-rhythms under different feeding paradigms
Supplementary Notes 2 and 3 show the modulation of the 24h-rhythms induced by 5 different feeding paradigms: 24 h ad libitum food access (AL); temporal restriction for 12 h during either the night (TR-night) or the day (TR-day); or 30% caloric restriction with 24 h access, starting at either the beginning of the night (CR-night) or the day (CR-day). These feeding paradigms imposed on mice produce dynamical perturbations of daily rhythmicity (Supplementary Figs. 10, 11). While the observed patterns are consistent with previous results[22], the advantage of the GaMoSEc approach is the ability to monitor daily changes in phase and strength of rhythmicity.

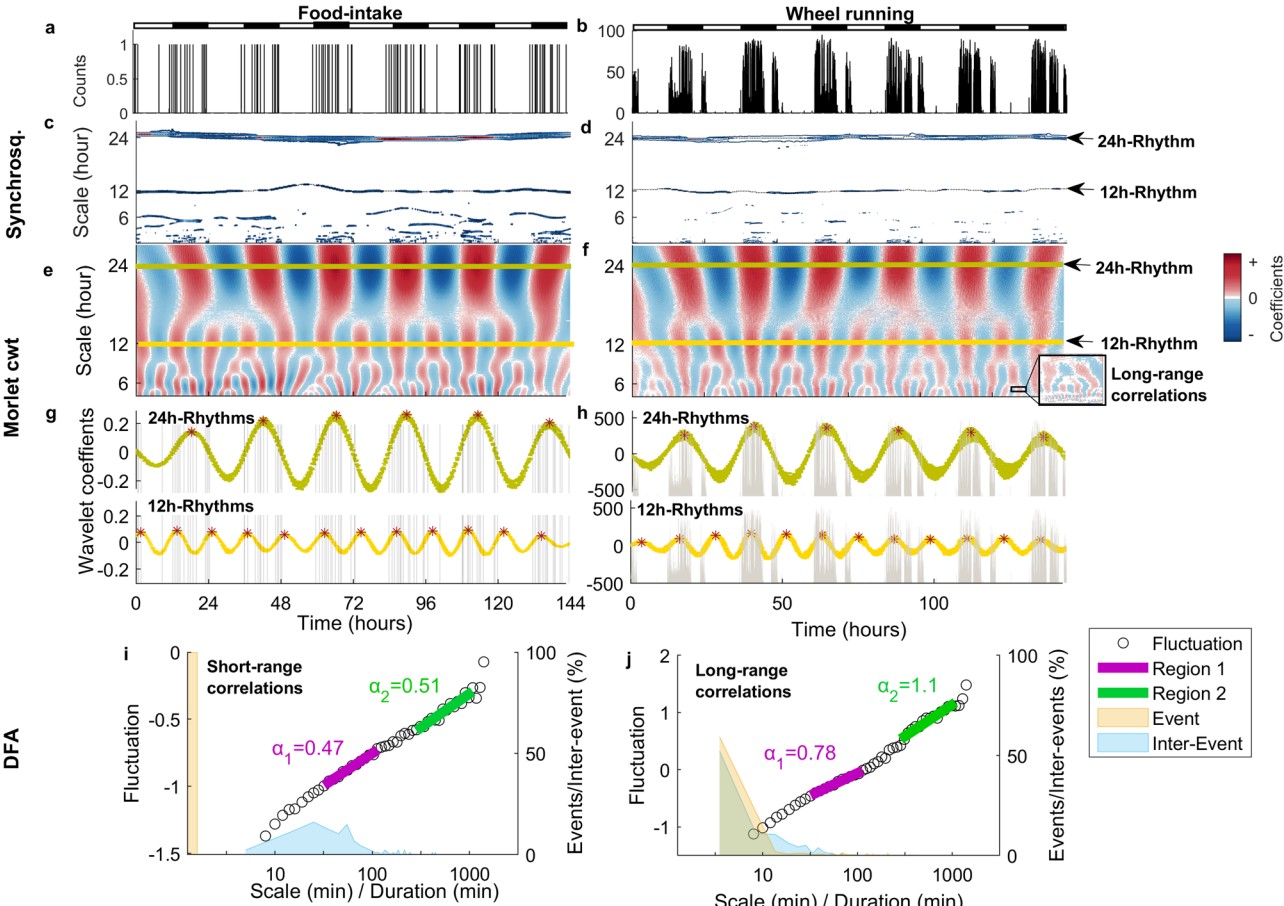

**Fig. 1 | 12h-rhythms and complex fractal dynamics in the time series of mice under the ad libitum (AL) feeding paradigm.** Examples of (**a**) food-intake and (**b**) wheel running time series. **c, d** Contour plot of the wavelet coefficients obtained with syncrosqueezing analysis of these time series, respectively. Positive coefficients are shown following the corresponding color scheme. Dotted lines indicate the first two strongest ridge detected connecting maximum coefficients, characterizing both the circadian rhythms and 12 h UR. **e, f** Coefficients of the Morlet Wavelet transform. Positive values are shown in red and negative values in blue, highlighting transitions between behavioral states (i.e., active vs. non-active) at each temporal scale (*y* axis) for the same two time series. Around the 24 h time scale (lime green line), the circadian rhythm is apparent (compare with **a, b**) as equidistant spaced blue and red bands. At smaller time scales, ultradian rhythms (around 12 h, yellow line) and complex fluctuations are visualized. **g, h** Examples of the circadian and 12 h ultradian rhythms found in time series (lime green and yellow lines, respectively) superimposed with the behavioral activity (underlying gray bars, same as **a, b**). **i, j** Detrended fluctuation analysis of time series on food intake (**i**) and wheel-running activity (**j**). Data correspond to animal F147[22] and code[86] are publicly available. The two α-values (see Table 1) of each individual were estimated as the slope of these curves in the two linear regions indicated with purple for $\alpha_1$ or green for $\alpha_2$. The observed α-values indicate long-range (fractal) correlations in wheel running but not in food-intake. Dual *y* axis, left represents fluctuations and right axis events (activity) and inter-events (rest between consecutive activities).

## Modulation of 12h-rhythms under different feeding paradigms

Since ultradian rhythm prevalence decreases with the rhythm period (Table 1), we limited our additional analysis of ultradian rhythms to the 12h-rhythms. Figure 2a–j show that 12h-rhythms are intermittently expressed (i.e, there are temporal intervals where 12h-rhythms are not detected) along the experiment and their expression pattern depends on behavior, individual background and feeding paradigm imposed.

### 12h-rhythm expression is independent between behaviors

Within the same individual, the presence of 12h-rhythm in one behavior does not ensure presence in the other behavior (Fig. 2a–j, see example: mouse 168 in AL group; mouse 146 in TR-night group; mouse 181 in TR-day group). Moreover, while the 12h-rhythms in food-intake are intermittently expressed, in wheel running activity they are more persistent along the whole experiment, which reveals specific behavioral differences in the underlying basis of regulation, interaction and/or production of 12h-rhythms. Note that the variability between animals is lower in wheel running than in food-intake activity (Fig. 2k, n), indicating that its expression is more consistent among individuals.

## Mice under the same feeding paradigm present strong inter-individual differences in the expression of 12h-rhythms

Inter-individual differences in the expression of 12h-rhythms are evident when we analyze mice under AL paradigm (Fig. 2a, f): intermittencies in 12h-rhythm expression are strongly observed in food-intake activity (Fig. 2a); whereas for the wheel running (Fig. 2f), half of the individuals present 12h-rhythms continuously throughout the whole experiment (Fig. 2f), while the other half present intermittencies. These intermittencies are not synchronized between animals, and, importantly, they seem not directly linked to genetic differences in timing of food consumption intrinsically associated with this strain of mice (see Supplementary Note 5 and ref. [22]).

In addition, we detect a slight increase in the number of mice exhibiting 12h-rhythms during the last week of the experiment as compared to the first week, especially when food-intake behavior is analyzed (i.e.: for AL paradigm, first week 0/6 individuals; last week 3/6 individuals). This effect on the proportion of individuals is not statistically significant ($p = 0.09$), but it may suggest that the length of the experiment may be an important factor for the consolidation of 12h-rhythms expression.

**Table 1 | Characterization of behavioral patterns in mice feed ad libitum under 12 L:12D cycle, during the first week of experimentation**

| Pattern type | Feature | Food-intake (n = 30) | Wheel running (n = 30) |
|---|---|---|---|
| ~24 h rhythms | Animals with rhythm (%)[a] | 100 | 100 |
| | Acrophase ZT[b] | 17.4 (16.8; 17.8) | 16.5 (16.1; 17.0) |
| | Correlation between animals[c] | 0.91 (0.87: 0.97) | 0.96 (0.94: 0.97) |
| ~12 h rhythms | Animals with rhythm (%)[a] | 17 | 93 |
| | First peak ZT[b] | 2.0 (1.7; 2.5) | 3.4 (3.0; 4.1) |
| | Second peak ZT[b] | 14.2 (13.9; 14.4) | 15.3 (15.1; 16.0) |
| | Correlation between animals[c] | 0.77 (0.70; 0.86) | 0.82 (0.73; 0.88) |
| ~8 h rhythms | Animals with rhythm (%)[a] | 20 | 20 |
| | First peak ZT[b] | 5.7 (5.6; 6.5) | 6.3 (6.2; 6.7) |
| | Second peak ZT[b] | 13.8 (13.7; 14.2) | 14.5 (14.3; '4.7) |
| | Third peak ZT[b] | 21.6 (21.5; 22.6) | 22.4 (22.2; 22.7) |
| | Correlation between animals[c] | 0.70 (0.41; 0.76) | 0.76 (0.68; 0.85) |
| ~6 h rhythms | Animals with rhythm (%)[a] | 0 | 20 |
| DFA | $\alpha_1$-value (first slope)[d] | 0.51 (0.50; 0.52) | 0.95 (0.90; 1.00) |
| | $\alpha_2$-value (second slope)[d] | 0.36 (0.28; 0.49) | 1.15 (1.02; 1.22) |

[a]Percent of animals where the rhythms were detected using the 5-step GaMoSEC procedure.
[b]Time of daily peak in the real Morlet cwt coefficients at the given time scale.
[c]All pair-wise comparisons in real Morlet cwt coefficients between the 30 animals, total 870 comparisons at the given time scale.
[d]Detrended Fluctuation Analysis was used to estimate the self-similarity parameter, $\alpha$, using a detrending order of 3. Time scales used for $\alpha_1$ and $\alpha_2$, are shown in Fig. 1 panels i and j.

## 12h-rhythms are sensitive to the feeding paradigm imposed

Despite the high inter-individual variability, the feeding paradigm imposed on mice significantly affects the 12h-rhythm expression, especially in food-intake behavior (Fig. 2a–j). Qualitative analysis of food-intake behavior indicates that TR paradigms tend to increase (Fig. 2b, c), while CR paradigms tend to decrease or abolish (Fig. 2d, e) the 12h-rhythm expression, as compared to AL (Fig. 2a). We further quantify, for each testing group, after feeding schedule change, the median number of days with detected 12h-rhythms (Fig. 2k) and confirmed a significatively increase under the TR-day treatment. In addition, we detected phase advance (Fig. 2l, see also Supplementary Fig. 15) and a slight tendency to increase the strength of 12h-rhythmicity in the TR-day as compared to the control AL group (Fig. 2m, see also Supplementary Fig. 16). For both CR paradigms, the median number of days with 12h-rhythm does not decrease significantly (Fig. 2k). However, note that by the end of the experiment both CR conditions do not exhibit 12h-rhythms in feeding behavior (Fig. 2d, e, see Supplementary Figs. 17 and 18 in Supplementary Note 6 for details in time evolution of phase and strength of 12h-rhythms along the experiment).

Contrary to that seen with food-intake behavior, wheel running in the CR-night group is the only group in which we detect changes in the expression of 12h-rhythms due to feeding paradigm changes. Note that intermittences were observed in every individual after feeding change introduction (Fig. 2o, Supplementary Note 6), resulting in a significantly lower median number of days with 12h-rhythm (Fig. 2n). Although the number of days did not significantly change under the TR paradigms, it is worth noting that group variability decreased, suggesting consolidation of this rhythm (Fig. 2o). Acrophases of 12h-rhythms in running were not affected by the new feeding paradigm introduced (Fig. 2o, see also

Supplementary Fig. 15), but the CR-day treatment significantly increases the strength of 12h-rhythms, as compared with AL condition (Fig. 2p, see also Supplementary Fig. 16).

Figure 2, panels q and r explore whether the 12h-rhythms could be related to the misalignment between feeding and running 24h-rhythms. We found that misalignment between feeding and running 24h-rhythms does not completely explain the differences found in the expression of 12h-rhythms between groups. This is clearer for the running time series: most of the groups (except CR-night), present a similar number of days with 12h-rhythms, irrespective of the feeding treatment (Fig. 2n, r). For the feeding time series, the expression of 12h-rhythms seems most related to the food consumption pattern (see Discussion), although a slight effect of the misalignment between 24 h behavioral rhythms cannot completely be ruled out, especially for the CR paradigm (Fig. 2q).

Taken together, these results support the hypothesis that the food restriction paradigm modulates the presence, phase and/or strength of 12h-rhythms. Misalignment between 24 h behavioral rhythms has little or no significant effect on their expression and modulation.

## Modulation of autocorrelation properties induced by the feeding paradigm

We further explore with DFA whether the autocorrelation properties are affected by the feeding paradigms imposed on mice (see Supplementary Fig. 14). Supplementary Note 7 shows the temporal evolution of autocorrelation parameters during the entire experiment. Figure 3a, b display representative examples of DFA performed on the behavioral time series from the last 5-days of testing for each feeding paradigm.

As shown previously, for the first week of testing (Fig. 1i, j), at least two scaling regions are apparent; hence, the self-similarity parameters, $\alpha_1$ and $\alpha_2$, can be used for group comparisons. For feeding time series, as in the first week (Table 1), AL presented $\alpha_1$-values near 0.5 indicating random or short-range correlations. However, TR-day, CR-night and CR-day present a decrease in $\alpha_1$-value in comparison to the AL group ($P < 0.05$, Fig. 3c). These lower $\alpha_1$-values between 0 and 0.5 indicate a change in the properties of the dynamics towards anti-correlation. Anti-correlation in this context can be associated with events of feeding being followed by prolonged non-feeding episodes. For TR-day, $\alpha_2$-values increased as compared to AL group remaining close to 0.5 (Fig. 3c). For CR, the $\alpha_2$-values increased above 1, reflecting the fact that under these conditions, long periods without any feeding activity (i.e., smoother time series[25].

For the wheel running time series, $\alpha_1$-values presented long-range correlations in all animals, while showing a tendency ($P < 0.1$, Fig. 3d) to be lower in TR-day and CR-night and were significantly lower in CR-day ($P < 0.05$, Fig. 3d) in comparison to AL mice. These lower $\alpha_1$-values indicate modulation of the structure of long-range correlations within the minute to several hour range. In regard to the larger time scales, no significant differences were detected between treatment groups, with $\alpha_2$-values predominantly between 1 and 1.5 (Fig. 3d).

Thus, feeding protocols differentially modulate the correlation properties of both behaviors depending on the time scale. For food-intake activity this modulation occurs for the two $\alpha$-values, in opposite ways, whereas for wheel running activity only $\alpha_1$-values were affected. Importantly, the $\alpha_1$-value reflects the fluctuations in activity dynamics at time scales shorter than the observed 12h-rhythms. Hence, the feeding paradigm not only changes rhythmic behavior but also modulates non-oscillatory fluctuations at these smaller time scales, affecting the scaling properties of behaviors.

## PCA highlights the role of dynamical patterns beyond 24h-rhythms for complete characterization of the feeding paradigms groups

Figure 4 shows a set of different multivariate principal component analysis (PCA) combining variables estimated for 24h-rhythms (acrophases), 12h-rhythms (number of days with 12h-rhythms) and autosimilarity parameter ($\alpha_1$-values) with other physiological variables previously reported for each mice group[22]. For comparison, pair-wise scatterplots (Supplementary

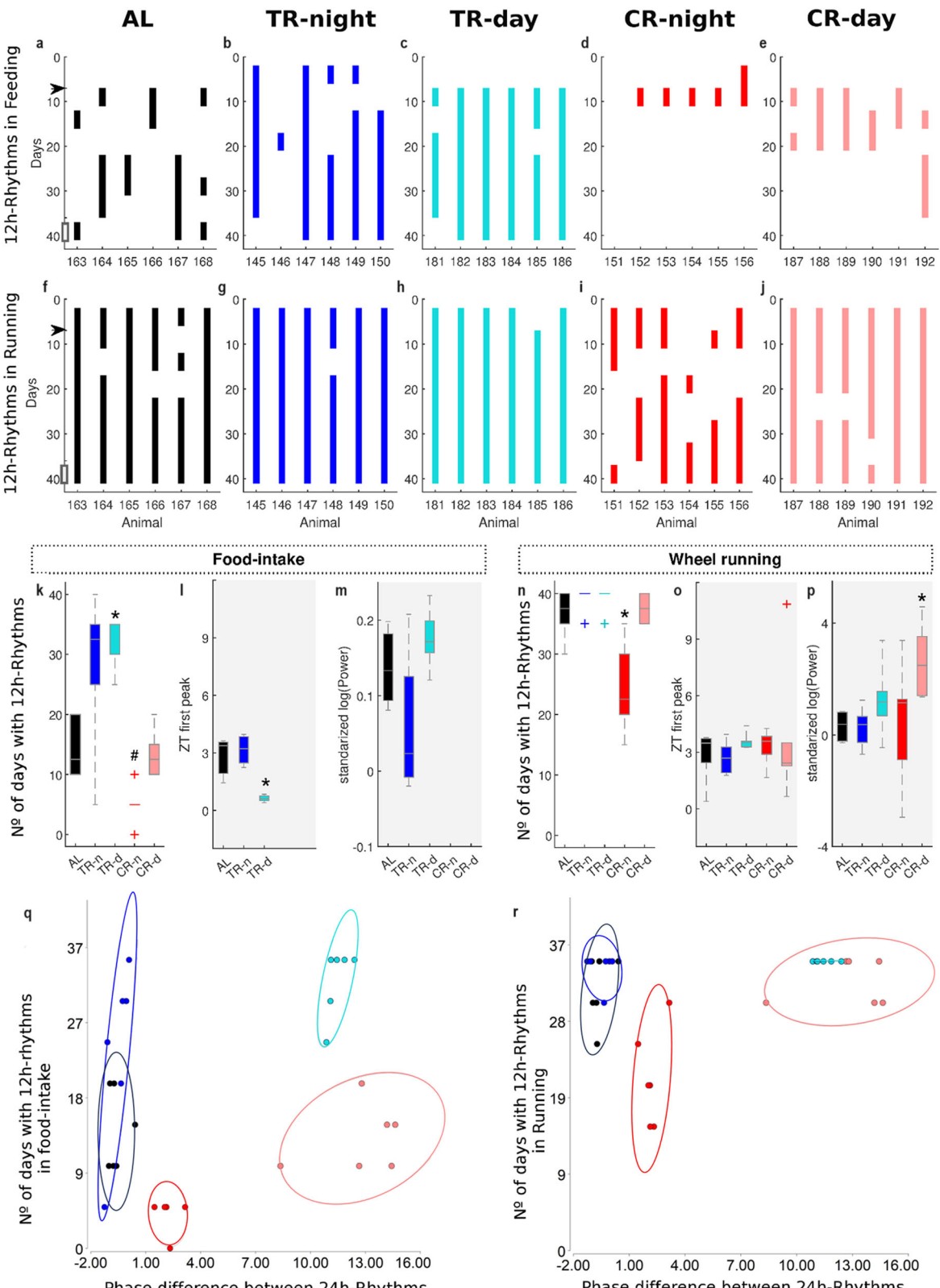

Fig. 20) and correlation estimation (Supplementary Fig. 21) between variables are shown in Supplementary Note 8.

PCA that included 24h-rhythm acrophases and number of days with presence of 12h-rhythms revealed that these variables were sufficient to separate CR-night from the rest of the groups. Nevertheless, they are insufficient to separate TR-day from CR-day (Fig. 4a). When $\alpha_1$-values are included (Fig. 4b), these groups were distinguishable among them. Further inspection denotes that the PC1, principally associated with $\alpha_1$-values and wheel running acrophase, divides CR to the left from TR and AL to the right. In addition, PC2 associated principally with circadian feeding acrophase (Supplementary Tables 1–4), allows separation of day and night groups (TR-day and CR-day versus TR-night and CR-night). Note that the

**Fig. 2 | Intermittent expression of food-intake and wheel running 12 h rhythms.** **a–j** Colored vertical bars indicate each five day period in which 12 h ultradian rhythms were detected in each animal studied in regards to **a–e** food-intake and **f–j** wheel running using the 5-step GaMoSEC wavelet approach (Supplementary Notes 1 and 2). Black arrow shows the moment of transition to the novel feeding paradigm. **a, f** AL: ad libitum paradigm. **b, g** TR-night (TR-n): 12 h food access during the night. **c, h** TR-day (TR-d): 12 h food access during the day. **d, i** CR-night (CR-n): 30% caloric restriction with 24 h food access starting at the beginning of the night. **e, j** CR-day (CR-d): 30% caloric restriction with 24 h food access starting at the beginning of the day. Statistical comparison of paradigms using a Kruskal–Wallis test of the values of the **k, n** number of days with ultradian rhythms (UR) ($H = 19.85$,

$p = 0.0004$ and $H = 13.6$, $p = 0.003$), as well as quantification of (**l, o**) first peak ($H = 9.42$, $p = 0.0007$ and $H = 6.36$ y $p = 0.17$) and (**m, p**) power in the 12-rhythms ($H = 4.83$, $p = 0.07$, and $H = 13.59$, $p = 0.009$) during the last 5-days of experimentation in (**k–m**) food-intake and (**n–p**) wheel running. Box plots in **k–p** show the quartiles 1,b2 and 3. Whisker depict the 5 and 95 percentiles. *Treatment group significantly differs from AL (P < 0.05). Power values were transformed using logarithm and then standardized to the baseline value (i.e., mean values obtained between days 2–5 of experimentation). Relationship between the phase difference between circadian food intake and wheel running behaviors and the number of days in which 12h-rhythms in **q** food-intake and **r** wheel running was detected. Circles indicate the confidence ellipse of 95% for each treatment group.

persistence of 12h-rhythms in both feeding and wheel running is also affecting the vertical separation between groups in an opposite way than 24h-rhythm feeding acrophase (Fig. 4b).

The addition of physiological variables (Fig. 4c, Supplementary Tables 5 and 6) only changes the intra-group variability; however, groups remain separate. PCA of the physiological variables by themselves (Fig. 4d), shows the largest intra-group variability, resulting in less capacity to discriminate between groups. Only PC1, principally associated with the variables body and stomach weight (Supplementary Tables 7 and 8), allows separation of the AL and TR-night groups from CR-day groups.

Combined, these results highlight the importance of considering dynamical patterns coexisting overall time scales (24h-rhythms + 12h-rhythms + autocorrelation properties) for the understanding of animal behaviors as an integral complex system, sensitive to modulation by external cues.

## Discussion

We characterize behavioral time series of mice by combining GaMoSEC and DFA. Our study reveals the concurrent impact of the imposed feeding paradigm on different types of behavioral patterns, encompassing rhythms and autocorrelations spanning across a wide-range of temporal scales. These results shed light on the underlying complex architecture of mice behavior. Complexity in this context refers to the presence of multi-scale dynamical patterns in time series of mice behaviors. Although these patterns are detectable, we still lack specific information about whether and how these temporal domains exchange information and mutually influence each other. Each dynamical domain (circadian, ultradian, and autocorrelations) could present specific susceptibility to perturbations by certain environmental signals (e.g., feeding protocols imposed on mice), which in turn could impact other scales. Keeping in mind complexity and the associated multi-scale dynamical patterns aids in constructing an integral, and thus realistic, understanding of animal behavior.

As previously reported[5], behavioral complexity is evident under control (ad libitum, AL) conditions (Table 1, Fig. 1). Mice not only exhibit ~24h-rhythms within the circadian domain but also ~ 12, ~8, and ~6-hour periods rhythms within the ultradian domain, along with short- or long-range autocorrelations. Particularly, the autocorrelation properties present in time series unveil the dynamical microstructure of behaviors and provide information on behavioral memory. Behavioral memory refers to the probability that a behavioral action at a given time point (i.e., running or eating events) strongly depends on previous events of the same behavior. Behavioral dynamical processes, described through time series, can exhibit positive (auto) correlations, anti-correlations or no correlations between behavioral time points. These reflect different types of behavioral dependencies over time, and, therefore, behavioral memory. Positive correlations indicate that an event in the present makes it more likely that the same event will occur in the future. Anti-correlations indicate that an event in the present makes it more unlikely that the same event will occur in the future. Correlations can also be characterized by the duration of these dependencies. Long-range (auto) correlations indicate that these temporal dependencies persist over several orders of temporal magnitude in time series. Thus, they can be mathematically associated with a power-law function (i.e., linear regions in Fig. 1i, j, characterized by a slope, $\alpha$), which presents poor

decay over time and is thus associated with long-term memory. Short-range (auto) correlations, on the contrary, imply fast temporal decay of temporal dependencies; therefore, the process can be considered as a short-term and mathematically associated with the quick exponential decay of correlations over time[25,26]. In this context, the difference between the short-range correlations ($\alpha = 0.5$) found in feeding and long-range correlations ($0.5 > a > 1$) seen in wheel running resides in how correlations persist. $a$-values above 1 and below 0.5 are indicative of persistent strong correlations (see example Supplementary Fig. 12a, c, e) or anti-correlations, respectively[18]. Since α-values represents the microstructure of behavioral dynamics and are unaffected by differences in the mean level of activity[10] they are often more sensitive to stress, aging and illnesses than traditional summary statistic measures (i.e., counts, means, etc).

Beginning in the circadian domain, our characterization of the ~24h-rhythms aligns with previous analysis of the same dataset[22]. Likewise, the detected periods are consistent with a prior study on locomotor behavior in mice and rats fed ad libitum[16]. However, herein we go beyond these previous findings: we quantified daily changes in phase and strength of 24h- and 12h-rhythms evolving throughout the whole experiment (Supplementary Figs. 10, 11, and the specific discussion in Supplementary Note 3).

Although ultradian rhythms have been previously reported in various animal species[3,27–32] at the organismal/physiological[17,33–41], tissue[42,43], and cellular levels[15,44–50], in mammals, their origin, functionality, modulation by external cues and interaction with other temporal domains are not yet fully understood[34,51–56]. Two different hypotheses have been proposed to explain the generation of ultradian rhythmicity. According to one hypothesis, ultradian rhythms arise from an internally dedicated ultradian clock, independent of both the circadian system and the photoperiod[33,41,52,57–59]. Supporting this view, a highly tunable dopaminergic ultradian oscillator (DUO), independent of the SCN was detected to drive ultradian locomotor rhythms in mice, with period lengths from a few hours to multiple days. Interestingly, DUO has also been associated with the ability for mice to entrain food-based zeitgebers[60]. Additionally, a cell-autonomous 12h pacemaker, independent from the circadian clock, was recently described in mouse liver, which regulates 12h rhythms of gene expression[59]. According to the other hypothesis, ultradian rhythms may be biological harmonics of circadian rhythms, emerging from superimposed circadian rhythms out of phase. This hypothesis has been used to explain ultradian gene expression in liver and other tissues[61–63]. Although the mechanism at a behavioral level is unclear, it possibly could involve neuronal network dissociation by conflicting external signals influencing the SCN neuronal coupling. This mechanism is consistent with the model postulating the existence of the E (evening) and M (morning) oscillators in the SCN of nocturnal rodents[64]. It also corresponds to the observed splitting phenomena in hamsters' locomotor activity, where the left and right sides of the SCN express TTFL genes in antiphase[65,66]. However, this last hypothesis implies that the generation of ultradian rhythms is necessarily dependent on the circadian system. Recently, ultradian rhythmicity was found to persist in the absence of functional molecular circadian clocks at both behavioral and cellular levels[15,17]. Of interest are the four types of ultradian oscillations detected in wheel running time series of Per1/2/3 KO mice[17], since two of them, with periods in the range of 14–20 h and between 5 and 8 h, agree with the rhythms we detected and characterize here (Table 1, Fig. 2). Moreover, these

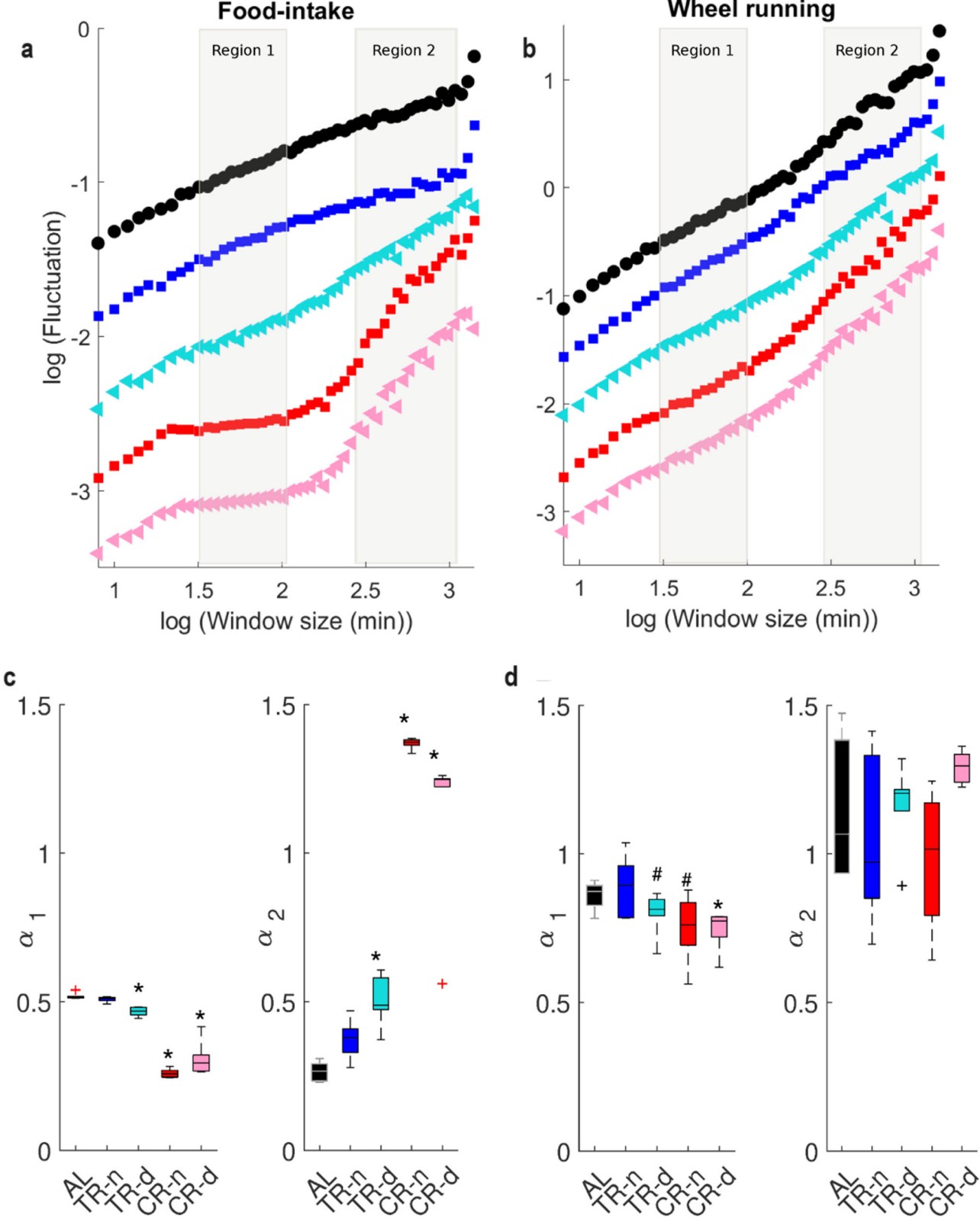

**Fig. 3 | The feeding paradigm modulates differentially autocorrelation properties of behavioral dynamics depending on the temporal scale.** Representative detrended fluctuation analysis of **a** food-intake and **b** wheel running behavioral time series (days 36–40) for each feeding paradigm. Gray background highlights the first scaling region, $\alpha_1$. Boxplot of $\alpha$-values estimated for **c** food-intake and **d** wheel running behavioral time series for each treatment group. Box plots in **c, d** show the quartiles 1, 2, and 3 and whisker depict the 5 and 95 percentiles. A Kruskal–Wallis test showed significant effects of feeding paradigm in $\alpha_1$ and $\alpha_2$ (see scaling regions marked in Fig. 1i, j) for food-intake ($H = 26.45$; $P < 0.0001$ and $H = 26.81$, $P = 0.03$, respectively) and in Fig. 1l in $\alpha_1$ for wheel running ($H = 10.69$; $P = 0.03$) *Treatment group is significantly different from the control AL group ($P < 0.05$). #Represents a tendency to differ from control $P = 0.09$. AL: ad libitum paradigm. TR-n: 12 h food access during the night. TR-d: 12 h food access during the day. CR-n: 30% caloric restriction with 24 h food access starting at the beginning of the night. CR-d: 30% caloric restriction with 24 h food access starting at the beginning of the day.

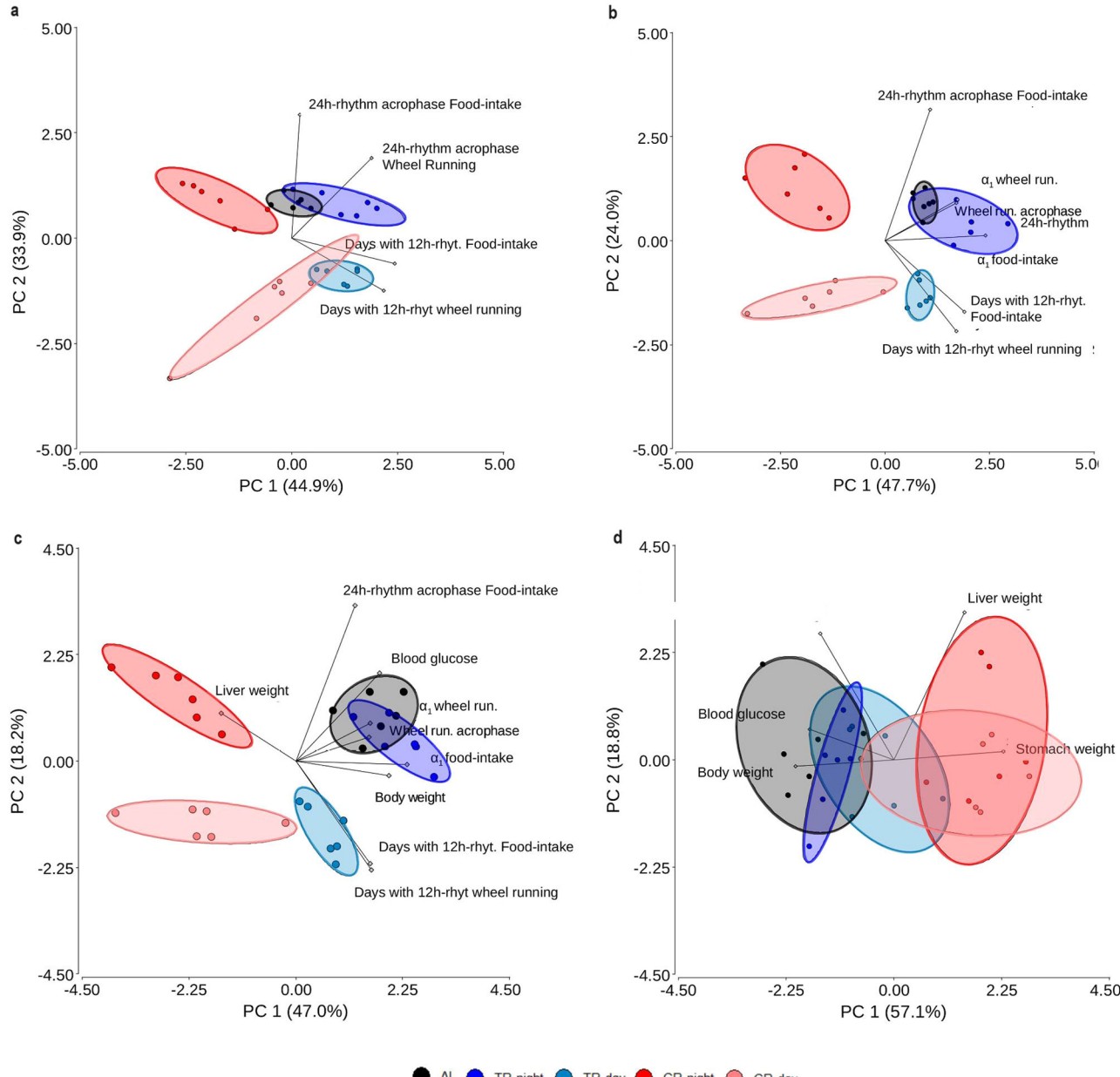

**Fig. 4 | Principal component analyses of mice under different feeding paradigms.** PCA that include behavior dynamics are included in the analysis (**a, b**) without and/ or (**c, d**) with inclusion of physiological variables reported in Acosta-Rodríguez et al.[22] from the final stages of the experiment. Each circle represents an individual, with the color denoting the treatment as indicated in the legend key. Food-intake and Wheel running acrophase associated with the circadian rhythm is the same as data shown in Fig. 2. The number of days in which 12h-rhythms were observed in each behavior is the same as shown in Fig. 2q and l, as with α-values in Fig. 3c, d.

two ultradian components detected in the Per1/2/3 KO mice present intermittently high and low amplitudes, resembling the intermittences we characterize here for the ~12h-rhythms. It should be kept in mind that these two coexisting hypotheses are not totally mutually exclusive, highlighting the need to continue research in this field.

We have previously modeled and contrasted these two hypotheses regarding the origin of ultradian rhythms from a wavelet analysis perspective[16] but have found limitations to distinguish between them. Therefore, we cannot definitively exclude either of these possibilities. Moreover, it is tempting to speculate that the rhythmic patterns detected in the ultradian domain may be a collection of rhythms/events produced by combination of these mechanisms. Since wavelet approaches such as GaMoSEC can simultaneously detect and characterize diversity of ultradian patterns, they promote insights for quantitative analysis in the field. This, in turn, sets the stages for future advancements toward the precise

identification of the mechanisms underlying ultradian rhythmicity. Another advantage of our wavelet approach is that the first of the five steps of GaMoSEC uses a Gaussian cwt, does not involve a prior assumption of a sinusoidal periodic pattern (Supplementary Figs. 1, 2 and 9). This is important since ultradian rhythms potentially can present a stochastic, jagged appearance[37,67]; even a non-sinusoidal, square, waveforms[15,17]. Detectability is not independent of the characteristics of ultradian rhythms (i.e., relative amplitude and shape)[37,67], thus appropriate methodological approaches must consider waveform[68]. In this regard, when periodicity is evident at the 12 h scale, changes in activity throughout the day occur smoothly with two peaks apparent during a 24h time span (Supplementary Fig. 10). Thus, the following four steps of GaMoSEC use sinusoidal-like functions for characterization. Note that the timeseries analyzed here are derived from mice with an intact circadian system, and therefore our wavelet analysis allows the contribution of sinusoidal signals to be considered.

Regarding ultradian rhythm, our observations also reveal intermittency in 12h-rhythmicity as well as strong inter-individual variability in persistence throughout the experiment (Fig. 2). This intermittency and variability observed at an individual level could help explain the previously reported low prevalence of these rhythms at the population level[18–20,52,54,56–58]. Notably, neither the presence of intact SCN, nor a consistent 12 L:12D cycle, guarantee 12h-rhythmicity over time (Fig. 2). To our knowledge there is no periodic external cue associated with the periodicities of ~8, and ~6 h ultradian rhythms detected in these experiments. Therefore, it is improbable that these ultradian rhythms are simply passive responses to external cues.

Even under control (AL) conditions, significant differences were observed between behaviors, particularly concerning both ~12-hour ultradian rhythms and autocorrelation properties. Specifically, ~12h-rhythms in wheel running are more prevalent and persistent compared to than those in food-intake time series. Furthermore, the ~12h-rhythms identified in both behaviors appear to be uncoupled: at a given time point, food-intake ~12h-rhythms may not be evident in some mice, while wheel running ~12h-rhythms persist (Fig. 2). Similarly, contrasting dynamics between behaviors are also observed in autocorrelation properties of mice feed ad libitum. Food-intake presents short-range correlations with a single α-value characterizing its dynamical microstructure ($a \sim 0.5$, Table 1, Fig. 3c). Contrarily, for wheel running two different α-values are observed: long-range correlations are observed for time scales up to ~100 s (Region1 1; $a_1 \approx 0.95$; Table 1), while a stronger correlation structure is observed for larger time scales (Region 2; $a_2 \approx 1.15$; Table 1). This phenomena of multifractality have been previously observed in other complex behaviors[69–72]. These dynamical differences between behaviors could be associated with the divergent underlying processes of each one as will be discussed below.

Behavior dynamics under ad libitum conditions should be understood contextualized within the protocol used for measurement. This can be highlighted by comparing our results with two prior similar studies. In the first place, the short-range correlations found in mice feed-intake observed in our study contrast with the long-range correlation in feeding behavior found previously in quail[16]. However, these studies are profoundly different, not only in regard to biological differences between species but also in the type of food (pellet vs finely grained fed) and the method of recording (number of pellets taken vs. time spent at feeding trough). Also, for mice, a lag of at least 10 min between pellets was used[22] which imposes a detection limit of dynamical microstructures of food-intake time series for timescales lower than 10 min. In second place, the muti-fractal wheel running dynamics observed herein contrasts with prior studies of locomotor activity of healthy rodents fed ad libutum which showed monofractal long-range correlation dynamics (i.e. characterized by a single scaling exponent, α)[5,18,73,74]. The differences between these two behaviors can be attributed to wheel running being a putative, incentive-induced motivated behavior in itself[75], and may interact with different motivational systems than general locomotor behavior[76]. Because access to a running wheel impacts both total food consumption and circadian rhythmicity[77]. Therefore, caution must be maintained when comparing the dynamical microestructure of different behaviors, remembering that time series are deeply influenced by the technique chosen to describe behavior. This caveat is generalizable to ultradian rhythms as well. Therefore, it will be interesting in the future to test the effect of different types of feeder settings and evaluate how the presence of a wheel impacts on the dynamical patterns of behaviors under ad libitum conditions.

A central finding in this work is that feeding schedules modulate both behavioral ultradian rhythms and dynamical microstructure. Regarding the ultradian domain, the decoupling of ~12 h rhythms between running and feeding within the same mouse, already observed under ad libitum conditions, can be further induced by changing the dietary schedule (Fig. 2). Note that daytime restriction (TR-day) paradigms tend to induce, while the caloric restriction (CR) paradigms tend to abolish the ~12h-rhythms in food-intake time series, while the ~12h-rhythms of wheel running persist under TR or diminish (under CR-night) the ~12h-rhythms. Differences in persistence of ~12h-rhythms between feeding protocols cannot be attributed to phase differences of 24h-rhythms (Fig. 2q, r) or to the total amount of food ingested (Supplementary Fig. 16 and see[22]). At the same time, the relative robustness of wheel running ~12h-rhythms to the feeding paradigm imposed resembles the resilience of the SCN against food restriction protocols[78,79]. This robustness is also in line with the subtle impact of these type of dietary conditions over circadian wheel running activity[78,79], and Supplementary Note 3. Nevertheless, GaMoSEC is sensitive enough to detect significant changes in the persistence (under CR-night paradigm, Fig. 2n), phase inter-individual variability (denoted as higher level of correlation between individuals, under TR-day paradigm, Supplementary Figs. 9i, 15h) and strength (under CR-day paradigm, Fig. 2p) of the ~12h-rhythms in wheel running time series. Together these results suggest a straightforward link in the modulatory pathway of feeding restriction protocols on 12h- food-intake rhythms.

Over time, feeding behavior under caloric restriction (CR) resulted in diminishing behavior complexity. This is evident not only in the loss of 12-hour rhythms but also as strong anti-correlations at short time scales ($a_1 < 0.5$) and as a dramatic increase in autocorrelations ($a_2 > 1$) for larger time scale in food-intake time series. We postulate that this loss of complexity is related to the food consumption pattern developed by the mice, which self-impose a 2h-temporal restriction (i.e., food is quickly consumed within 2 hours[22]). Note that the consolidation of feeding occurs more rapidly under CR-night than under CR-day, which is evident from the first days after feeding paradigm change (Supplementary Figs. 10 and see[22]), coinciding with the complete disappearance of 12h-rhythms (Supplementary Fig. 17) and a sharp decrease in $a_1$-values under CR-night (Supplementary Fig. 19). The slower consolidation of CR-day is also reflected in all domains (Supplementary Figs. 10, 17, 19 and see[33]), as a more gradual phase shift in ~24h-ryhms, rate of disappearance ~12h-rhythms of food-intake and changes in α-values. These observations regarding the speed of consolidation in food-intake behavior indicate a dynamical link between all the temporal domains.

The phase shift in the 12h-rhythms of food-intake under TR-day and in CR-day (before they completely disappear (Supplementary Fig. 17), was significant when compared with their night counterparts (Supplementary Fig. 15b–e). Also, under TR-day the feeding behavior presents a decrease in phase dispersion between animals as compared with TR-night (Supplementary Fig. 15b–e). When 24h-rhythms in feeding and wheel running occur in opposite phases of the day (i.e., TR-day and CR-day), the blood glucose increases when the mice are eating[22]. Interestingly, metabolic stressors, specifically the excess or deprivation of glucose, synchronize 12h rhythms in vitro[53]. Together these observations suggest that the glucose profile, mainly driven by the feeding/fasting cycles, could be a potential zeitgeber of 12h-rhythms of feeding behavior in vivo.

Acosta-Rodríguez et al.[22], had previously shown that both CR groups exhibited an increase in daytime wheel running activity (but not % of overall 24 h activity) by the end of the study, which reflects the food anticipatory activity (FAA) of mice. In CR-night the nocturnal activity transiently advanced (days 14–19) into the middle of the rest phase[22]. We show that this altered dynamical pattern was evident on all domains (see Supplementary Figs. 10, 17 and 19), decreasing the persistence of the 12h rhythm (Fig. 2i, n) and increasing the randomness of the autocorrelations, evidenced as a decrease of the $a_1$-values (Fig. 3d). In contrast, CR-day showed a fast transition towards heightened behavioral complexity. In comparison to ad libitum controls, the strength of 12h-ultradian rhythms increased (Fig. 2p), but $a_1$-values also became less correlated (Fig. 3d, i.e. more random), although remaining in the realm of long-range correlations. Therefore, FAA is not sufficient to explain observed effects of CR on the ultradian domain of running dynamics, given the described contrast between CR-night and CR-day. Thus, the link between the FAA and the ultradian domain requires further exploration. However, or results supports previous evidence showing that food restriction protocols (i.e., 2-h food availability during the daytime) increase randomness of the autocorrelations properties in locomotor activity of rats in comparison to ad libitum controls[19].

It is currently unknown whether signals from peripheral clocks specifically related to the metabolic status of mice (i.e. brown and white adipose tissues, skeletal muscle and liver) are involved in the generation/modulation of the observed dynamics. These tissues express high levels of 12 h cycling transcripts[80,81] and as previously mentioned, a cell-autonomous mammalian 12h pacemaker has been described in mouse liver[62]. Recently, ultradian rhythms from locomotor activity in voles were related to their energy balance and metabolic status[35]. Similarly, ultradian rhythms in cell cultures were only observed when cultures were at confluence, which seems to be a key condition to achieve energy balance through gap junction-mediated coupling[15]. In depth studies are required for elucidating how food restriction protocols relate to energy balance and the expression of ultradian rhythms in both running and feeding mice behaviors. In addition, the link between circadian dynamics and autocorrelation properties has been thoroughly documented in rat locomotor activity[82,83], although not in food-intake nor wheel running. The SCN has been proposed to act as a major neural control node that is central for the generation of long-range correlation of locomotor activity at multiple time scales[18,78,82], particularly relevant for time scales between 4 and 24 h[18]. This is also consistent with our results in animals with an intact SCN where no significant effects of the feeding paradigm were observed in $\alpha_2$-values. However, we also observed significantly affect $\alpha_1$ supporting the hypothesis of Lo et al.[19], that a neural network of coupled multiple control nodes is responsible for scale-invariance. This includes another neuro-anatomical source(s) (other than the SCN) that are fundamental for shorter time range. More recent studies include as a node of this network the dorso-medial hypothalamic nucleus involved in the food anticipatory activity. Other nodes could be dopaminergic ultradian oscillator (DUO), as well as the liver and/or other peripheral tissues. Environmental contexts, such as the feeding paradigm would have modulatory effect on this network leading to profound physiological changes. This contention is consistent with multivariate analysis that combines behavioral dynamics and physiology variables (Fig. 4c). Four clusters are evident, roughly distributed in each corner of the PCA plots. While ad libitum and TR-night groups overlap in the top-right corner the other groups are completely separated into the other three corners. Interestingly, CR-day is found in the opposite, bottom-left, corner of the PCA plot representing the profound contrast both at a behavioral as well as physiological level to ad libitum and TR-night. Although no strong level of correlations can be observed between individual variables ($R^2 < 0.58$; Supplementary Fig. 21), $\alpha_1$ shows a significant correlation with each physiological variables highlighting the importance of loss of behavioral complexity in feed-intake in this complex system. Additionally, the power to separate groups introduced by including $\alpha$-values (compare Fig. 4 panels a and 4b) in the PCA analysis can be associated with its sensitivity to defect profound dynamical changes in time series. A recent study that followed mice under CR feeding protocols throughout their life has shown that, both CR-day and CR-night, are effective at increasing lifespan as compared to ad libitum[84]. Similarly, CR protocols tend to decrease $\alpha_1$ values in wheel running time series as compared to ad libitum. Thus, our results not only support a multimode control network but also raise questions regarding the relationship between behavioral complexity and aging.

In all, our work contributes to highlighting the complex nature of mouse behavior. Here, we take a step forward to provide a more specific sense of the sometimes fuzzy word complex. We offer quantitative tools that allow us to precisely define what we mean when we think about complexity in the temporal domains of mouse behavior. Moreover, the way in which each behavior integrally responds to a given perturbation (in this case, feeding restriction protocols), is not commonly studied. Our work addresses this gap by demonstrating how food restriction (both TR and CR) involves a dynamic reconfiguration of the underlying physiological networks associated with each behavior (feeding and running), altering their temporal complexity, likely through different internal mechanisms. This, in turn, sets the stages for future advancements toward the precise identification of the mechanisms underlying the dynamical patterns (rhythms and autocorrelation structure) and the links that build the architecture of these complex behaviors.

## Methods
### General procedure
A detailed description of experimental protocol is provided in Acosta-Rodríguez et al.[22]. Briefly, C57BL/6 J male mice (8 weeks old, n = 30) were individually housed in standard cages equipped with a customized automatic feeder. Water was provided ad libitum and a 12:12 LD cycle was used throughout the study. Mice were fed with round pellets of 315 ± 4 mg each containing 3.35 Kcal/g. ClockLab Chamber Control Software was used to program feeding schedules and record food-intake events. A 10 min delay was programmed after each pellet was taken before the next pellet was dropped to prevent hoarding behavior. After one week of recording under ad libitum food access, mice were randomly assigned to one of 5 feeding conditions ($n = 6$ per condition): 24 h access ad libitum (AL); unlimited amount but temporally restricted to 12 h during the dark (TR-night), or light (TR-day) phase; and 24 h access but calorically restricted (11 pellets corresponding to 70% of baseline ad libitum levels) fed at the start of the dark (CR-night) or light (CR-day) phase. Animals were maintained under these conditions for the following 35 days; thus the total experimental period was 42 days. The cages were changed on day 21. Number of pellets (i.e., food-intake) and number of revolutions of the activity wheel (i.e., wheel running) were recorded throughout the study using a 1 min sampling interval. Given that the feed pellet dispenser recorded feeding as discrete events, prior to analysis, time series were smoothed using a moving average function with a 1 h moving window[5]. Data analysis was performed between day 1 and 42, thus providing a 12 h habituation period to the novel environment. The source of data[22] stated that all animal protocols were approved by the Institutional Animal Care and Use Committee (IACUC) of the University of Texas Southwestern Medical Center (APN 2015-100925).

### Five-step wavelet approach GaMoSEC for behavioral time series analysis
The GaMoSEC, 5-step wavelet approach was implemented for analysis of behavioral time series (see details in Supplementary Note 1 and 2 and refs. [5,16]).

Detection and characterization of circadian and ultradian rhythms were performed using a combination of different wavelet decomposition techniques that simultaneously detrend and denoise the signal. Wavelets have the potential to describe the data without making any parametric assumptions about trends (i.e., changes in mean value of the signal over time) in the frequency or amplitude of the components signals and are resilient to noise (see review in ref. [5]). Also, information regarding changes in temporal dynamics over the length of the experiment at different time scales is quantifiable. Hence it is possible to detect the consolidation or disappearance of a given ultradian rhythm. It is noteworthy that wavelet analysis is not a single analysis but rather a family of analyses defined by the characteristics of the wavelet used in the transformation (i.e., Gaussian, or Morlet continuous wavelet transform). Herein, the time series data were consecutively analyzed step-by-step with the transformations described below, each highlighting different aspects of data.

First step: visual inspection by Continuous wavelet transform based on a real Gaussian mother wavelet in the Cartesian time scale plane. This wavelet transform highlights changes in the signal and singularities (i.e., spike-like or step-like changes) in the dynamics of the time series[5].

Second step: visual inspection by Continuous wavelet transform based on complex Morlet mother wavelet in the polar time scale plane. This is a complex wavelet of periodic nature, thus its transformation is also complex, providing 4 different plots corresponding to the real, imaginary, modulus, and phase angle of the wavelet coefficients. This complex wavelet provides information regarding the presence of oscillatory behavior, and herein the real part is also used to estimate the acrophase[5].

Third step: modal frequency identification by Empirical wavelet transform, which is a wavelet analysis in the Fourier domain followed by frequency segmentation to extract the modal components. This is an independent analysis that can also detect rhythms in time series, as well as changes in periodicity.

Fourth step: modal frequency identification by Synchrosqueezed wavelet transform, a linear timescale analysis followed by a synchrosqueezing technique. This analysis provides highly localized frequency information, important for precise estimation of period and power of rhythms[5].

The two behavioral time series (food-intake and wheel running) of each animal were analyzed using these four types of transformations. Results were then divided into 5-day segments for graphical representation and statistical analysis. Initially, for characterization of normal ad libiitum conditions we focused on only the first week of testing (day 1–6). After-wards, the modulation of behavior due to changes in the feeding paradigm was assessed, and focus was placed on the last week of testing (days 36–40) given that the period was observed to be stable over time. At each of these two time periods evidence of sustained rhythms were evaluated. For each animal, if evidence of periodicity was observed in all 4 analyses at a specific time scale (i.e. 24 or 12 h) and during a specific time period, then the given rhythm was considered to be detected under the specified experimental condition (see details of criteria in Supplementary Note 2). Percent of animals with the given rhythm was estimated.

Fifth step: quantification of coherence and phase difference between different series, providing important information regarding phase relationships between signals.

### Quantification of rhythmic behavior

From the quantification of the first 4-steps of GaMoSECs described in the previous section, the following variables were obtained (code used in ana-lyses is publicly available[85], see also Supplementary Note 3 for details): period of detected rhythms was estimated from the localization of the ridge of the synchrosqueezing algorithm (black dotted lines in Fig. 1c, d); power of the detected rhythm was estimated as the squared modulus of the complex syncrosqueezing wavelet coefficients along the ridge (Fig. 1c, d); Acrophase was estimated as the hour of the day in which the maximum values of the real part of the complex Morlet continuous wavelet transform were observed (Fig. 1g, h, maximums marked with red stars); correlation between animals for both behaviors (i.e. food-intake or wheel running) a correlation coefficient was estimated between the real Morlet wavelet coefficients of each pair of animals for the 24 h and 12 h time scale (see schematic repre-sentation in Supplementary Fig. 6 and details in ref. [16]); phase shift between behavioral time series was estimated using the wavelet coherence algorithm (Supplementary Figs. 7–9).

### Quantification of autocorrelations in behavioral time series using DFA

The method[25] utilized herein to determine scale-invariance and to evaluate the presence and extent of long-range autocorrelations in food-intake and wheel-running activity, was described in Supplementary Note 4. Briefly, DFA estimates the self-similarity parameter, $\alpha$, that measures the auto-correlation structure of the time series. If $\alpha = 0.5$, the series is uncorrelated (random) or has short-range correlations (i.e. the correlations decay exponentially), whereas $0.5 < \alpha < 1$ indicates that long-range autocorrela-tion exists (correlation decays as a power-law), meaning that present depends on past behavior[25]. Also, $\alpha$ is inversely related to a typical fractal dimension, so in this case, the $\alpha$-value increases with increasing regularity in the time series. This software is also available in the public domain (http://www.physionet.org/physiotools/dfa/). Herein, DFA calculations were per-formed with a customized script ran on MATLAB R2018a.

Trends within the behavioral time series were also systematically studied[26]. A DFA of third order was the lowest detrending order that elimi-nated trends in all series, and therefore it was applied to all series for esti-mating $\alpha$-values. In addition, the appropriate scaling range was determined using the following criteria: stable values of local slopes, maximum coefficient of variation, and minimum sum of squared residuals (Supplementary Fig. 13 for details[73]). This analysis showed a stable scaling region for all food-intake and wheel running for time series between 10–100 min, range used for $\alpha_1$. A second scaling region, $\alpha_2$, was established between 285–1176 min.

### Statistics and reproducibility

The dataset analyzed herein has been previously published[22], thus a sample size of 6 was determined previously for the original study. The effects of the feeding paradigm on variables associated with behavioral dynamics were evaluated using a Kruskal–Wallis nonparametric test since overall data did not comply with the assumptions of normality and homogeneity of variance (checked by Shapiro–Wilk test and Fisher's $F$ test, respectively) using InfoStat; level of significance for the rejection of the null hypothesis was set at $p < 0.05$. Statistical data is presented as box plots.

### Reporting summary

Further information on research design is available in the Nature Portfolio Reporting Summary linked to this article.

### Data availability

The dataset analyzed herein has been previously published[22] and is publicly available on Mendeley (https://doi.org/10.17632/hxwwyycjy7.1). The source data behind graphs in Figs. 1–3 can be found in Supplementary Data 1.

### Code availability

Customized script in MATLAB is publicly available on FigShare for GAMoSEC[85] (figshare https://doi.org/10.6084/m9.figshare.21545385), and for DFA[86] (https://doi.org/10.6084/m9.figshare.1514975.v1). The Empirical Wavelet Transform Matlab toolbox is freely distributed on MATLAB Central File Exchange (https://www.mathworks.com/matlabcentral/fileexchange/42141-empirical-wavelet-transforms).

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

## Acknowledgements

We thank Drs. Fransico Tamarit, Sergio A. Cannas, and Carlos A. Condat for constructive feedback, useful comments, and suggestions. J.M.K., P.S.N., and A.G.F. are career members of CONICET. J.S.T. and V.A.A.-R. have grant support from R01 AG045795/AG/NIA NIH HHS/United States, K99 GM132557/GM/NIGMS NIH HHS/United States, HHMI/Howard Hughes Medical Institute/United States, R56 AG072736/AG/NIA NIH HHS/United States and R35 GM127122/GM/NIGMS NIH HHS/United States.

## Author contributions

J.M.K., A.G.F., P.S.N.: resources, investigation, conceptualization, data analysis, review, editing; J.M.K, P.S:N.: writing original draft, figure creation; V.A.A-R: conceptualization, review, editing. J.S.T: review, editing. All authors participated in review and editing of previous versions, and also read and approved the final manuscript.

## Competing interests

The authors declare no competing interests.
