## [Peer Review File · Communications Biology]

Reviewers' comments:

Reviewer #1 (Remarks to the Author):

The authors present a complex spectral analysis of mouse behavior under a variety of feeding regimens. This study uses data from a previously published study (Acosta-Rodríguez et al., 2017, *Cell Metabolism* 26, 267–277

July 5, 2017). With the analyses present the authors detect a number of ultradian rhythms, and focus their analyses on a 12h rhythm which comes and goes under the different feeding regimens.

While the analyses are quite sophisticated and detailed, the presentation of the manuscript is very hard to follow. The results and discussion are very long-winded and unfocused, making it hard to follow the arguments presented. A more concise presentation would enhance the overall impact of the manuscript. Inclusion of some actograms from the original study would assist the reader in understanding the data that were analyzed. Since those are open access and the original authors are authors on this paper, inclusion of the actograms should be straight-forward.

My largest concern overall is around the claim that the 12h rhythm is internally generated (line 415-418). The problem here is that many aspects of the original study could contribute to detection of a 12h rhythm when there really isn't an internal 12h timer. A 12h rhythm is a harmonic of the 24h circadian rhythm. Also, Mice frequently show a bimodal activity pattern with a major activity bout at dusk and a secondary bout prior to dawn. Since the LD cycle used here was LD12:12, this would lead to these bouts being ~12h apart, contributing to detection of a 12h rhythm that might not be observed under longer or shorter photoperiods. The appearance and disappearance of the rhythms under the various feeding paradigms is not actually that surprising. In the calorie restricted animals, once the food is gone, there can be no more feeding, so the loss of the 12h feeding rhythm is unsurprising. If the authors take a thoughtful approach about the source of the 12h rhythms, and then focus on the phenomenon and how the spectral analysis informs our understanding of the phenomenon, then I think this sort of analyses will be more impactful for the field.

Reviewer #2 (Remarks to the Author):

In this MS, Kembro et al used signal processing method to analyze mouse feeding and wheel-running behaviors under different feeding regimens. They specifically focused on ~12h ultradian rhythms as well as fractal dynamics (autocorrelation) in the time series data. Overall, this is a well-executed study with rigorous analysis and lengthy discussion (maybe a little but too long) on the potential implications of their findings. The manuscript is well-written and largely easy to follow, leaving very technical descriptions to supplemental material. This study is a follow-up of their recent work published in Scientific Reports, and of considerable importance and interest to the field of chronobiology, specifically to the field of ultradian rhythms.

One of the major difficulties in studying ultradian rhythms is that they are often superimposed by circadian rhythm with larger amplitude and other ultradian oscillations, and thus harder to detect. Therefore, new mathematical and engineering tools that differ significantly from traditional statistics-based methods, like JTK-CYLE, RAIN, Metacycle are needed. In this sense, this study provides another good example of how to use integrated Wavelet analysis to uncover ultradian rhythms. The DFA revealed another layer of complexity in the fractal feature of mouse behaviors, and how different feeding behaviors may alter these features.

While the whole study is very technically sound, this reviewer thinks in its current form, it lacks sufficient integration between rhythm features and biological features, namely whether and (how) different features of rhythms in feeding and wheel running is correlated/associated with biological features such as body weight and blood glucose in each mouse. After all, this paper is submitted to Communications Biology (not signal processing or an engineering journal), and the audience would be interested in knowing whether these rhythmic features are related to measurable physiological readout of health or disease. For instance, are individual parameters of 24h and/or 12h rhythm phase/persistence/strength, a_1 and a_2 values correlated with body weight, blood glucose or any other biological readouts for each mouse, within or even across different feeding regimens? If not, are PC1 values depicted in 4A or 4B correlated with these features? In the discussion section, the authors cited a few papers on how fractal features in HRV are associated with health. What about the fractal features of feeding behaviors and wheel-running activity? Are they also indicators of health in mice?

Some minor comments are also presented below.

1. Line 133, it would be useful to also include the age and sex information of the mice used for analysis.
2. Line 191, 30% CR with 24h access is not technically correct, as mice with CR self-imposed a TRF, and food are quickly taken. I think the language should reflect this nuance.
3. Discussion is really long, and it would be useful to have subheadings to break up the discussion.
4. Figure 1. Please put a light/dark label on top as well.
5. Figure 3. Panel E, F are not labeled.
6. Figure 4, it would be useful to add scatter plot/PCA plot to show how different feeding groups segregate with only 24h arcophase feeding and wheel-running information, only 12h persistence information, and only a_1 and a_2 information.

Reviewer #3 (Remarks to the Author):

This paper builds on excellent scholarship published elsewhere by the manuscript's various authors to harnesses a 5-step wavelet technique and detrended fluctuation analysis to reanalyze previously published work by Acosta-Rodríguez et al. 2017. The dataset analyzed used male c57Bl6 mice whose locomotor activity and food intake were monitored longitudinally for 42 days. Subgroups of these mice experienced various temporal or caloric food restriction after the 1st week of ad libitum feeding. Compared to the circadian focus of the previous report, the authors here seek to understand other neglected, but important elements of the temporal structure of the behavior and physiological processes under study. They specifically want to understand the data through the lens of a complex system. By the end of the current manuscript, they extensively quantify how non-circadian temporal features of locomotor activity and food intake available in their data set are altered by temporal and caloric restriction.

The question the authors seek to to address, is in my opinion, important, as is trying to understand the temporal features of behavior through new lens not as well used by the circadian focused field at large. As the authors note: ultradian (<24 hr) rhythms or events, depending on your interpretation of recent results, are circadian independent and fairly ubiquitous through out nature in various behavioral and physiological traits, but poorly understood. Documented by reviews from AD Grant and colleagues (2018) and Prendergast and Zucker (2016), there remains a large discrepancy between the number of circadian and ultradian focused papers published. This is driven in a large part by a couple of factors. Ultradian rhythms/events are nonstationary and don't manifest consistently across the circadian cycle, making traditional circadian analysis techniques based in fourier methods unsuitable for their analysis and in many cases making them difficult to consistently discern in raw time series. Additionally unlike other biological rhythms, ultradian rhythms are not obviously synced to an external periodicity like the day or year as is the case of the circadian and seasonal counterparts. This latter fact has limited our ability to determine the physiological substrates from which they originate, and how even basic manipulations affect their manifestation. As such the chronobiology field has struggled with their investigation. Work by Tanya Leise and Mary Harrington in 2011 suggested that wavelet analysis might be the way to study these ultradian components, but much scholarship including some of the authors' own in this last year has been focused on refining these techniques and beginning to address basic questions about the properties of these ultradian rhythms. As the authors also note: to date, the putative origins of ultradian rhythms/events remain undetermined, though work by Blum and colleagues (2014) suggest that a dopamine mechanism may be at play. Additionally recent scholarship from Flores et al 2016 suggests that there may be vital links between the oscillator identified by Blum and colleagues and the ability for animals to entrain to food based zeitgeibers. The authors' goal of understanding the question of how temporal features of mouse locomotor activity and food intake (beyond circadian rhythms) are affected by temporal and caloric restriction of food then, is of interest both to the chronobiological field and other fields of biology.

Overall, I am pretty convinced by the quality of the analysis within the paper. The changes observable at the level of the basic behavior would suggest that there are non circadian changes occurring in the

temporal structure of feeding and locomotion when mice experience food restriction of various sorts. It is nice to see it formally characterized at last. I have qualms, however, with: the accessibility of its writing at times, and more significantly with the data set used in pursuit of this analysis. Its limitations are not adequately treated with in the current manuscript, which is a shame because in my opinion the manuscript has far more interesting implications when taken in the light of these caveats. I have some specific concerns/comments about the manuscript in its current form.

1. "Auto correlation properties" in the abstract may be fairly unparsable for the general reader. Can this be restated slightly? Specifically through a more biological focused lens.

2. Likewise the introduction could benefit from reorganization and clearer bridging between mathematical concepts and what they reflect about the biology. One of the larger challenges in communicating work like this is despite its importance is that the less computationally inclined reader can quickly become overwhelmed and lose track of the point. Having thoroughly read and thought about the paper, the introduction is now much more understandable, than in my naive reading but there are several places where the authors could increase the accessibility of their work by "connecting the dots." Some connections I don't think are very clear are (1) what is the overall problem/question the paper is seeking to address, (2) what gap thinking about ultradian rhythms as a complex system bridges and (3) what detrended fluctuation analysis is and why understanding the correlation properties of the system is important beyond being another descriptor.

3. Please specify that the results are from exclusively male mice as there are several documented sex differences in food intake and ultradian rhythms (see various scholarship from the laboratories of Lance Kriegsfeld, Brian Prendergast, and Arthur Arnold).

4 Citations numbering doesn't always appear to align with the reference list. For example on line 683 in the Methods: Peng et al. is attributed to citation [90] but is actually [98] in the reference list. I assume this is probably an error with some citations becoming unlinked in the authors' reference manager. It would be worth verifying the reference #s are all properly linked to their intended citation.

5. The supplementary text for example (lines 136-137) doesn't seem to always align with the supplementary figures or the text is perhaps unclear when the author's discuss rows and columns. Given how much information is communicated in the supplementary methods, ensuring their clarity will go a long way to making the work much more reproducible.

6. Can the authors further comment at all in their analysis' ability to distinguish between 12 h rhythms that may be tied to changes in SCN coupling dynamics i.e. the proverbial E and M oscillators and those that are ultradian in the sense that they are generated by a mechanism other than that which is generated in the SCN? I know they make minor comment to some of this effect already (lines 411-415), my gut instinct would be that these fluctuating 12 h rhythms are more consistent with the output of changes in coupling dynamics rather than being an independent oscillation, but this is a pretty subtle but important point for future scholarship that is not currently elucidated on in the manuscript.

7. The data set the authors have chosen to use to address their questions has some limitations that the authors do not sufficiently treat with that affect the interpretability of their results. As the authors themselves note the way food intake is measured may affect their results (lines 547-549) but the use of running wheels is also not neutral, see scholarship by (Reebs, S. G., & Mrosovsky, N. (1989). Effects of induced wheel running on the circadian activity rhythms of Syrian hamsters: entrainment and phase response curve. *Journal of biological rhythms*, 4(1), 39-48. AND Edgar, D. M., Martin, C. E., & Dement, W. C. (1991). Activity feedback to the mammalian circadian pacemaker: influence on observed measures of rhythm period length. *Journal of biological rhythms*, 6(3), 185-199.) for further discussion. Importantly for the authors' purposes running wheels visually changes the microstructure of locomotor activity. Mice under passive infrared detectors or with telemetry implants exhibit far less consolidated rhythms with markedly increased diurnal activity compared to animals on a running wheel. These animals all still exhibit robust circadian rhythms, but observationally probably have quite different dynamics occurring in the ultradian range. This is a pretty visually striking difference. Given that both their measurement techniques impose structure on the system whose structure they are studying are the authors concerned at all about that important properties of the system might be obscured i.e. less ultradian rhythms or ultradian components masked by harmonics of the circadian rhythm especially given the rhythms the authors detect are where you'd expect to find harmonic i.e. 12 h, 8 hr, 6 hr? In a perfect world I'd love to see the same data collected with more neutral measurement techniques, but in lieu of that can the authors reinterpret their data with appropriate caveats or provide additional evidence why this is not of concern?

Reviewer #1 (Remarks to the Author):

1. The authors present a complex spectral analysis of mouse behavior under a variety of feeding regimens. This study uses data from a previously published study (Acosta-Rodriguez et al., 2017, Cell Metabolism 26, 267–277 July 5, 2017). With the analyses present the authors detect a number of ultradian rhythms, and focus their analyses on a 12h rhythm which comes and goes under the different feeding regimens. While the analyses are quite sophisticated and detailed, the presentation of the manuscript is very hard to follow. The results and discussion are very long-winded and unfocused, making it hard to follow the arguments presented. A more concise presentation would enhance the overall impact of the manuscript. Inclusion of some actograms from the original study would assist the reader in understanding the data that were analyzed. Since those are open access and the original authors are authors on this paper, inclusion of the actograms should be straight-forward.

Re: We appreciate the reviewers positive and constructive feedback. The revised version has been considerably shortened. Effort has been made to streamline the discussion section as well as incorporate the relevant concerns presented by all reviewers. Moreover, while we acknowledge the usefulness of actograms for readers, we chose not to include them in Figure 1 to maintain the clarity and focus of the message we intended to convey. Instead, we have incorporated explanatory text (Line 128) to direct readers to the actograms presented in Supplementary Figure 8, in Supplementary Section 3.

2. “My largest concern overall is around the claim that the 12h rhythm is internally generates (line 415-418). The problem here is that many aspects of the original study could contribute to detection of a 12h rhythm when there really isn’t an internal 12h timer. A 12h rhythm is a harmonic of the 24h circadian rhythm. Also, Mice frequently show a bimodal activity pattern with a major activity bout at dusk and a secondary bout prior to dawn. Since the LD cycle used here was LD12:12, this would lead to these bouts being ~12h apart, contributing to detection of a 12h rhythm that might not be observed under longer or shorter photoperiods”.

Re: This is an excellent point and has been reflected upon extensively in this revised version (lines 389-431). We agree that this study cannot resolve the debate on the origin of the observed rhythms, and it was not our intention to do so. That being said, our detailed study did show that the continuous presence of the 12L:12D cycle during the whole experiment and an intact SCN is not sufficient to induce ~12 h rhythms in food-intake or wheel running

consistently in individual mouse. Rather, as shown in Figure 2, these rhythms present intermittencies. We also informed the presence of ~8h- and ~6h-rhythms (Table1), although we did not dive into their characterization. To our knowledge these rhythms do not correspond to any known external cue/zeitgeber present in these experiments. Nevertheless, we cannot rule out some contribution of the LD cycle on the phenomenology of rhythms/events in the ultradian domain and recognize that it would be interesting to further investigate the contribution of a zeitgeber and/or masking cue on the ultradian behavior, all of which is out of the scope in this manuscript (lines 454-459).

We would also like to note that due to the general audience of the journal, we intentionally avoided the term “harmonic” in the previous version of our manuscript, given that can be used differently depending on the field of research. In chronobiological signal processing, it is common practice to identify a single (circadian) rhythm and deem higher frequency signals as mathematical harmonics, rather than resolving them as a second (or more) coexpressed rhythm" (van der Veen & Gerkema, 2017 doi: 10.1096/fj.201600872R). We would like to the reviewer note that in the GaMoSEC, the first step consists of Gaussian cwt, critically important since it is based on a wavelet function that is not in itself periodic, thus fundamental for understanding data variability over different time scales while ruling out spurious mathematical harmonics. This topic was already extensively discussed in our recent work (Flesia et al., 2022. Meth Mol Biol 2399: 277-341), and discussed in more detail in lines 436-449. Nevertheless, there is another meaning associated to the harmonics term, that we indeed have included in the new version of our manuscript (line 404). According to one hypothesis about ultradian rhythms generation, they may emerge from superimposed circadian rhythms out of phase, thus constituting *biological harmonics* of circadian rhythms. This hypothesis has been used to explain ultradian gene expression in liver and other tissues. Although the mechanism at a behavioral level is unclear, it possibly could involve neuronal network dissociation by conflicting external signals influencing the SCN neuronal coupling. Please note that we have included this debate in the new manuscript version (lines: 403-425).

3. “The appearance and disappearance of the rhythms under the various feeding paradigms is not actually that surprising. In the calorie restricted animals, once the food is gone, there can be no more feeding, so the loss of the 12h feeding rhythm is unsurprising.”

Re: Indeed, it is true that in calorie restricted animals end up having a self-imposed ~2h-temporal restriction for eating, which is obviously incompatible with the notion of 12h feeding rhythms. As discussed on Lines 527-543 in regard to CR feeding dynamics, the loss of 12h-

rhythms is not a passive consequence of food deprivation, but a more complex mechanism related to the food consumption pattern progressively developed by the mice, evident with the characterization we propose. Note that kinetics vary depending on when (day or night) the food is made available (Supplementary Figures 17 and 18). Thus, in this work we have advanced in the quantification of the temporal evolution towards behavioral reconfigurations.

4. “If the authors take a thoughtful approach about the source of the 12h rhythms, and then focus on the phenomenon and how the spectral analysis informs our understanding of the phenomenon, then I think this sort of analyses will be more impactful for the field.”

Re: We have thoroughly revised this manuscript, placing effort into conveying a thoughtful approach about the source of the ~12h rhythms (lines 389-431). Moreover, we place focus on how our methodological approach allows quantification of dynamical patterns over a broad range of temporal scales to include rhythms of different periods (i.e. ~24, ~12, ~8, etc) as well as auto-correlations (lines 363-380). In this way we take a step forward to provide a more specific sense to the sometimes "fuzzy" word "complex" (lines 351-362). We have also included specific statements on how our wavelet analysis can inform our understanding of the phenomenon (Lines 432-449); setting the stage for future advancements toward the precise identification of the mechanisms underlying the dynamical patterns (rhythms and autocorrelation structure) and their links.

Reviewer #2 (Remarks to the Author):

In this MS, Kembro et al used signal processing method to analyze mouse feeding and wheel-running behaviors under different feeding regimens. They specifically focused on ~12h ultradian rhythms as well as fractal dynamics (autocorrelation) in the time series data. Overall, this is a well-executed study with rigorous analysis and lengthy discussion (maybe a little but too long) on the potential implications of their findings. The manuscript is well-written and largely easy to follow, leaving very technical descriptions to supplemental material. This study is a follow-up of their recent work published in Scientific Reports, and of considerable importance and interest to the field of chronobiology, specifically to the field of ultradian rhythms.

Re: We genuinely appreciate the positive and constructive comments provided by the

reviewer. We believe that these insights have significantly contributed to enhancing the quality of our manuscript.

One of the major difficulties in studying ultradian rhythms is that they are often superimposed by circadian rhythm with larger amplitude and other ultradian oscillations, and thus harder to detect. Therefore, new mathematical and engineering tools that differ significantly from traditional statistics-based methods, like JTK-CYCLE, RAIN, Metacycle are needed. In this sense, this study provides another good example of how to use integrated Wavelet analysis to uncover ultradian rhythms. The DFA revealed another layer of complexity in the fractal feature of mouse behaviors, and how different feeding behaviors may alter these features.

Re: Thank you for this comment. In the discussion we have further highlighted how, by combining GaMoSEC with DFA, we are able to gain a more in-depth, quantitative, understanding of the complexity of mouse behavior (lines 432-449).

“While the whole study is very technically sound, this reviewer thinks in its current form, it lacks sufficient integration between rhythm features and biological features, namely whether and (how) different features of rhythms in feeding and wheel running is correlated/associated with biological features such as body weight and blood glucose in each mouse. After all, this paper is submitted to Communications Biology (not signal processing or an engineering journal), and the audience would be interested in knowing whether these rhythmic features are related to measurable physiological readout of health or disease. For instance, are individual parameters of 24h and/or 12h rhythm phase/persistence/strength, α_1 and α_2 values correlated with body weight, blood glucose or any other biological readouts for each mouse, within or even across different feeding regimens? If not, are PC1 values depicted in 4A or 4B correlated with these features? In the discussion section, the authors cited a few papers on how fractal features in HRV are associated with health. What about the fractal features of feeding behaviors and wheel-running activity? Are they also indicators of health in mice?”

Re: In this revised version we have put special efforts to integrate our findings regarding temporal dynamics with biological features (lines 587-624). To this end, we have included a new figure (Supplementary Figure 21) in Supplementary Section 8 showing the correlations between each parameter extracted from time series analysis (24h and/or 12h rhythm phase/persistence/strength and α_1 and α_2 slopes) of each mouse and its biological features

(body weight, blood glucose, stomach weigh, liver weight and WAT weight). The discussion has also been completely reorganized to highlight the relationship between rhythms, auto-correlations and physiology. Moreover, correlations between PC1 and PC2 values and features presented in Figure 4a-d, are now presented in Supplementary Tables 2, 4, 6, and 8, respectively. A discussion on the relationship between behavioral patterns and longevity has been added (lines 618-624).

Some minor comments are also presented below.

1. *Line 133, it would be useful to also include the age and sex information of the mice used for analysis.*

Re: We have included this information as suggested in line 643 of the subsection, General Procedure, in the Methods section.

2. *Line 191, 30% CR with 24h access is not technically correct, as mice with CR self-imposed a TRF, and food are quickly taken. I think the language should reflect this nuance.*

Re: In this revised version of our manuscript we have stressed that these groups presents 2h-self-imposed TRF in the discussion section (lines 531-538). However, we maintained the phrase “30% caloric restriction with 24 hr access starting at either the beginning of the night (CR-night) or the day (CR-day)” when we introduce for the first time the CR groups (line 185) to avoid misconception in regards to the original paper (Acosta-Rodríguez, et al., 2017) [21].

3. *Discussion is really long, and it would be useful to have subheadings to break up the discussion.*

Re: We have made a concerted effort to shorten the manuscript and streamline the discussion, addressing all concerns raised by the reviewers. It is of note that subheadings in the discussion section are not allowed in Comm Biology Journal.

4. *Figure 1. Please put a light/dark label on top as well.*

Re: We have included the Light/Dark label on top of Figure 1 as the reviewer request

5. *Figure 3. Panel E, F are not labeled.*

Re: We have included labels to panels in Figure 3.

6. ***Figure 4, it would be useful to add scatter plot/PCA plot to show how different feeding groups segregate with only 24h acrophase feeding and wheel-running information, only 12h persistence information, and only α_1 and α_2 information.***

Re: As the reviewer suggests, we have included a new figure (Supplementary Figures S20 in Supplementary Section 8) with only 24h acrophase feeding and wheel-running information, only 12h persistence information, and only α_1 and α_2 information to complement Figure 4.

Reviewer #3 (Remarks to the Author):

“This paper builds on excellent scholarship published elsewhere by the manuscript's various authors to harnesses a 5-step wavelet technique and detrended fluctuation analysis to reanalyze previously published work by Acosta-Rodríguez et al. 2017. The dataset analyzed used male c57Bl6 mice whose locomotor activity and food intake were monitored longitudinally for 42 days. Subgroups of these mice experienced various temporal or caloric food restriction after the 1st week of ad libitum feeding. Compared to the circadian focus of the previous report, the authors here seek to understand other neglected, but important elements of the temporal structure of the behavior and physiological processes under study. They specifically want to understand the data through the lens of a complex system. By the end of the current manuscript, they extensively quantify how non-circadian temporal features of locomotor activity and food intake available in their data set are altered by temporal and caloric restriction.

The question the authors seek to to address, is in my opinion, important, as is trying to understand the temporal features of behavior through new lens not as well used by the circadian focused field at large. As the authors note: ultradian (<24 hr) rhythms or events, depending on your interpretation of recent results, are circadian independent and fairly ubiquitous through out nature in various behavioral and physiological traits, but poorly understood. Documented by reviews from AD Grant and colleagues (2018) and Prendergast and Zucker (2016), there remains a large discrepancy between the number of circadian and ultradian focused papers published. This is driven in a large part by a couple of factors. Ultradian rhythms/events are nonstationary and don't manifest consistently across the circadian cycle, making traditional circadian analysis techniques based in fourier methods unsuitable for their analysis and in many cases making them difficult to consistently discern in raw time series. Additionally unlike other biological rhythms, ultradian rhythms are not obviously synced to an external periodicity like the day or year as is the case of the circadian and seasonal counterparts. This latter fact has limited our ability to determine the physiological substrates from which they originate, and how even basic manipulations affect their manifestation. As such the chronobiology field has struggled with their investigation. Work by Tanya Leise and Mary Harrington in 2011 suggested that wavelet analysis might be the way to study these ultradian components, but much scholarship including some of the authors' own in this last year has been focused on refining these techniques and beginning to address basic questions about the properties of these ultradian rhythms. As the

authors also note: to date, the putative origins of ultradian rhythms/events remain undetermined, though work by Blum and colleagues (2014) suggest that a dopamine mechanism may be at play. Additionally recent scholarship from Flores et al 2016 suggests that there may be vital links between the oscillator identified by Blum and colleagues and the ability for animals to entrain to food based zeitgeibers. The authors' goal of understanding the question of how temporal features of mouse locomotor activity and food intake (beyond circadian rhythms) are affected by temporal and caloric restriction of food then, is of interest both to the chronobiological field and other fields of biology”

Re: We thank the reviewer for their highly valuable comments and constructive criticisms. We value the depth, quality, and kindness with which they formulated their feedback. They have helped us improve the quality of our manuscript. In this revised version we have included reference to Flores et al, 2016 in our discussion section (lines 399-400) in order to highlight the potential links *between DUO and the ability for animals to entrain to food based zeitgeibers.*

“Overall, I am pretty convinced by the quality of the analysis within the paper. The changes observable at the level of the basic behavior would suggest that there are non circadian changes occurring in the temporal structure of feeding and locomotion when mice experience food restriction of various sorts. It is nice to see it formally characterized at last. I have qualms, however, with: the accessibility of its writing at times, and more significantly with the data set used in pursuit of this analysis. Its limitations are not adequately treated with in the current manuscript, which is a shame because in my opinion the manuscript has far more interesting implications when taken in the light of these caveats.”

Re: We have placed effort into improving writing, to make it more accessible to the broad audience of the Journal. We have also included in the discussion limitations of the dataset used in lines 478-503

I have some specific concerns/comments about the manuscript it its current form.

1. *"Auto correlation properties" in the abstract may be fairly unparsable for the general reader. Can this be restated slightly? Specifically through a more biological focused lens.*

Re: We modified the phrase in the abstract, it now reads “*the dynamical microstructure of behavior (i.e., autocorrelations properties)*” (lines 34-35). We have also rearranged the discussion to place a more biological focus regarding autocorrelation properties (lines 355-372 & 563-610).

2. *“Likewise the introduction could benefit from reorganization and clearer bridging between mathematical concepts and what they reflect about the biology. One of the larger challenges in communicating work like this is despite its importance is that the less computationally inclined reader can quickly become overwhelmed and lose track of the point. Having thoroughly read and thought about the paper, the introduction is now much more understandable, than in my naive reading but there are several places where the authors could increase the accessibility of their work by “connecting the dots.” Some connections I don’t think are very clear are (1) what is the overall problem/question the paper is seeking to address, (2) what gap thinking about ultradian rhythms as a complex system bridges and (3) what detrended fluctuation analysis is and why understanding the correlation properties of the system is important beyond being another descriptor.”*

Re: Thank you, tis comment is very helpful and we have worked to connect the dots following recommendation, both in the introduction (lines 79-106) and Discussion (lines 348-380) sections.

3. *“Please specify that the results are from exclusively male mice as there are several documented sex differences in food intake and ultradian rhythms (see various scholarship from the laboratories of Lance Kriegsfeld, Brian Prendergast, and Arthur Arnold).”*

Re: We have included the age and sex information of the mice used for analysis in line 643.

4 *“Citations numbering doesn’t always appear to align with the reference list. For example on line 683 in the Methods: Peng et al. is attributed to citation [90] but is actually [98] in the reference list. I assume this is probably an error with some citations becoming unlinked in the authors’ reference manager. It would be worth verifying the reference #s are all properly linked to their intended citation.”*

Re: We have checked citation numbering, and it is corrected in this revised version of this manuscript.

5. *“The supplementary text for example (lines 136-137) doesn't seem to always align with the supplementary figures or the text is perhaps unclear when the author's discuss rows and columns. Given how much information is communicated in the supplementary methods, ensuring their clarity will go a long way to making the work much more reproducible.”*

Re: We have thoroughly reviewed citations to supplementary figures and text for consistency. All references to rows and columns have been changed to indicate panel with the appropriate letter.

6. *“Can the authors further comment at all in their analysis' ability to distinguish between 12 h rhythms that may be tied to changes in SCN coupling dynamics i.e. the proverbial E and M oscillators and those that are ultradian in the sense that they are generated by a mechanism other than that which is generated in the SCN? I know they make minor comment to some of this effect already (lines 411-415), my gut instinct would be that these fluctuating 12 h rhythms are more consistent with the output of changes in coupling dynamics rather than being an independent oscillation, but this is a pretty subtle but important point for future scholarship that is not currently elucidated on in the manuscript.”*

Re: This is a very interesting point. Thus, we have included in the discussion section the debate about ultradian rhythms generation (lines 393-431).

7. *“The data set the authors have chosen to use to address their questions has some limitations that the authors do not sufficiently treat with that affect the interpretability of their results. As the authors themselves note the way food intake is measured may affect their results (lines 547-549) but the use of running wheels is also not neutral, see scholarship by (Reebs, S. G., & Mrosovsky, N. (1989). Effects of induced wheel running on the circadian activity rhythms of Syrian hamsters: entrainment and phase response curve. Journal of biological rhythms, 4(1), 39-48. AND Edgar, D. M., Martin, C. E., & Dement, W. C. (1991). Activity feedback to the mammalian circadian pacemaker: influence on observed measures of rhythm period length. Journal of biological rhythms, 6(3), 185-199.) for further discussion. Importantly for the authors' purposes running wheels visually changes the microstructure of locomotor activity. Mice under passive infrared detectors or with telemetry implants exhibit far less consolidated rhythms with markedly increased diurnal activity compared to animals*

on a running wheel. These animals all still exhibit robust circadian rhythms, but observationally probably have quite different dynamics occurring in the ultradian range. This is a pretty visually striking difference. Given that both their measurement techniques impose structure on the system whose structure they are studying are the authors concerned at all about that important properties of the system might be obscured i.e. less ultradian rhythms or ultradian components masked by harmonics of the circadian rhythm especially given the rhythms the authors detect are where you'd expect to find harmonic i.e. 12 h, 8 hr, 6 hr? In a perfect world I'd love to see the same data collected with more neutral measurement techniques, but in lieu of that can the authors reinterpret their data with appropriate caveats or provide additional evidence why this is not of concern?"

Re: We now addressed explicitly how the way food intake and activity is measured can affect their results (lines 478-503). The caveats about harmonics mentioned by the reviewer are also reflected in lines 403-431 and see our reply to reviewer #1, item 2.

REVIEWERS' COMMENTS:

Reviewer #1 (Remarks to the Author):

The revisions have improved the manuscript. There are a few typos to fix, but otherwise I'm satisfied. Beautiful work!

Line 99 – shorter instead of shorted

Line 375 – alpha vs alfa

Line 1097 - H=, p= missing values?

Line 1115 missing H=?

Reviewer #2 (Remarks to the Author):

The authors have satisfactorily addressed my concerns. However, I do notice a mistake in line 619-621 of the discussion section.

"Concomitantly, a recent study that followed mice under these feeding protocols throughout their life has shown that CR-day is the most effective at increasing lifespan". I believe the authors are referring to the Science paper (Acosta-Rodriguez, 2022, Science). In figure 2A of this study, clearly CR-night is more effective than CR-day in extending the lifespan. The authors need to correct this mistake in this discussion before publication. They need to properly cite this study as well.

Reviewer #3 (Remarks to the Author):

Overall I think the authors have improved their manuscript since the 1st revision. I do think however, that it could still benefit from a bit more revision in the form of how the data is presented. It is important to illustrate what the data here in can and cannot tell us, the authors really haven't demonstrated much about the ultradian system (seperable from the ciradian system) in particular, largely I suspect due to the way the use of running wheels and the 10 minute feeding schedule impose structure on the system as they note in the discussion. There is a paper out of the Prendergast Lab which shows you can actually get more accurate estimates of underlying ultradian structure by splitting the data into the light and dark component which the authors may find helpful if they wish to expand that aspect of the paper, but regardless most of the ultradian work focuses on 12 h rhythms which in the current study are impossible to separate from the circadian system and the autocorrelational properties. This is not an indictment of the study, of course, understanding how the system is being altered is important regardless if the structures at play originate from a separate ultradian system, which based on the available literature exists in some form, or is a byproduct of the circadian oscillator state. The crux of the matter though is that most of what is important about this work rests in the autocorrelation data, as the presence or absence of the 12 hour rhythm is not investigated in enough detail to give that much more more nuance about what is occurring in the underlying system when food cues and light cues are misaligned. It is thus imperative, if the study is to be elevated beyond "the use of fancy math demonstrates we can quantify differences where visually we knew there were differences" and is of use for scientists who want to build on this important work, that the authors communicate their data in such a way that the reader gets an intuitive sense of what the differences they are measuring mean. This is where the manuscript in its current form falls short

in my opinion. I don't think there's anything that makes me question whether the authors are measuring real or important differences, but even with further explanation I find it challenging to really understand what more "Long term memory" in the system really implies about the unknown mechanisms driving this phenomena. As such I have a few suggestions which I think may help the authors communicate their work:

-I find the initial paragraph of the introduction (49-58) a bit convoluted. If you've read the paper it is clear what the authors intent is, but if you are approaching the paper for the 1st time or having forgotten large amount of details about it, then it is very unanchored from any important context.

- Figure 1: DFA is super important to understand for the purpose of the paper, but it is not used commonly in biological rhythms research. I suspect the audience who'd be most interested in this work are chronobiologists and as such I think the authors could strengthen their manuscript by accommodating for the lack of familiarity. Figure 1 I and J are hard to interpret without further context. I don't think readers are going to have an intuitive sense of how to interpret these graphs in a meaningful way, especially because they are on different scales. Of course there is the supplementary information, but while it provides a lot of technical details of how DFA works, I don't think it or the results text really help the reader who is not a DFA expert or in the process of becoming one "get it". Riggle et al. 2023 recently did somewhat similar work and used a strategy of creating artificial timeseries with known temporal components and performing their analyses on these as way to give their analyses context. I wonder if a similar approach would work well here. Creating artificial time series that look like real data, but have a a high level of autocorrelation or low I think could provide good contrast to understand the real data and should be minimal work to produce. Additionally I think the color choice of the event/inter event makes it hard to parse on these figures.

-While I applaud the authors for explaining this concept of autocorrelations (363-380). I think it would benefit from another pass. I spotted both a typo "alfa" instead of the greek letter and I think an example of behavioral memory and and further explanation could be warranted.

-There is a typo on line 478: should be dynamics instead of dynamical

-It might be helpful to provide a short definition/example of anti-correlations

-The latter half of discussion 503-638 is great and is a good model for the authors to use elsewhere in the paper.

Reviewer #1 (Remarks to the Author):

The revisions have improved the manuscript. There are a few typos to fix, but otherwise I'm satisfied. Beautiful work!

Line 99 – shorter instead of shorted

Line 375 – alpha vs alfa

Line 1097 - H=, p= missing values?

Line 1115 missing H=?

Rsp: Thank you very much for your feedback. All typos have been corrected and H and p values included.

Reviewer #2 (Remarks to the Author):

The authors have satisfactorily addressed my concerns. However, I do notice a mistake in line 619-621 of the discussion section.

"Concomitantly, a recent study that followed mice under these feeding protocols throughout their life has shown that CR-day is the most effective at increasing lifespan". I believe the authors are referring to the Science paper (Acosta-Rodriguez, 2022, Science). In figure 2A of this study, clearly CR-night is more effective than CR-day in extending the lifespan. The authors need to correct this mistake in this discussion before publication. They need to properly cite this study as well.

Rsp: We thank the reviewer for their thorough revision of the ms. We have corrected this text as follows (lines 627-630):

"A recent study that followed mice under CR feeding protocols throughout their life has shown that, both CR-day and CR-night, are effective at increasing lifespan as compared to *ad libitum* [86]. Similarly, CR protocols tend to decrease α_1 values in wheel running time series as compared to *ad libitum*."

Reviewer #3 (Remarks to the Author):

Overall I think the authors have improved their manuscript since the 1st revision. I do think however, that it could still benefit from a bit more revision in the form of how the data is presented. It is important to illustrate what the data here in can and cannot tell us, the authors really haven't demonstrated much about the ultradian system (seperable from the circadian system) in particular, largely I suspect due to the way the use of running wheels and the 10 minute feeding schedule impose structure on the system as they note in the discussion.

RSP: Thank you for your comments. We have made an effort to improve how the data is

presented. As you pointed out, the limitations of the data analyzed were already included in the discussion.

There is a paper out of the Prendergast Lab which shows you can actually get more accurate estimates of underlying ultradian structure by splitting the data into the light and dark component which the authors may find helpful if they wish to expand that aspect of the paper, but regardless most of the ultradian work focuses on 12 h rhythms which in the current study are impossible to separate from the circadian system and the autocorrelational properties. This is not an indictment of the study, of course, understanding how the system is being altered is important regardless if the structures at play originate from a separate ultradian system, which based on the available literature exists in some form, or is a byproduct of the circadian oscillator state. The crux of the matter though is that most of what is important about this work rests in the autocorrelation data, as the presence or absence of the 12 hour rhythm is not investigated in enough detail to give that much more nuance about what is occurring in the underlying system when food cues and light cues are misaligned. It is thus imperative, if the study is to be elevated beyond "the use of fancy math demonstrates we can quantify differences where visually we knew there were differences" and is of use for scientists who want to build on this important work, that the authors communicate their data in such a way that the reader gets an intuitive sense of what the differences they are measuring mean. This is where the manuscript in its current form falls short in my opinion. I don't think there's anything that makes me question whether the authors are measuring real or important differences, but even with further explanation I find it challenging to really understand what more "Long term memory" in the system really implies about the unknown mechanisms driving this phenomena.

Rsp: We have made a considerable effort to improve communication of our data and to offer a more intuitive understanding of the significance of the observed differences, as elaborated in the responses below. We agree with the reviewer on the importance of our findings concerning the modulation of autocorrelation properties. However, we consider that we have also made a substantial contribution by emphasizing the complex nature of mouse behavior across a wide range of temporal scales. This is demonstrated by illustrating the impact of changes in an external cue on diverse behavioral dynamical patterns, encompassing biological rhythms and the dynamical micro-structure revealed by DFA. This underscores the necessity of adopting an integrated perspective when analyzing behaviors. Moreover, we provide quantitative tools that allow us to precisely define what we mean when we think about complexity in the temporal domains of mouse behavior. These tools are important to quantify otherwise qualitative observations. Therefore, we appreciate and welcome the valuable comments and suggestions provided by this reviewer below.

As such I have a few suggestions which I think may help the authors communicate their work:

-I find the initial paragraph of the introduction (49-58) a bit convoluted. If you've read the paper it is clear what the authors intent is, but if you are approaching the paper for the 1st time or having forgotten large amount of details about it, then it is very unanchored from any important context.

Rsp: We have worked to simplify the text. Additionally, we modified the first introductory paragraph to better contextualize the definition of a complex system within the broader scope of the paper. The paragraph reads as follows (lines 49-56):

"Animal behaviors can be conceptualized as part of a complex system. Such systems, found widely in nature from neuroscience to economics, share simple defining features: a large number of elements that interact with each other through non-linear relationships. These systems involve events and information flowing across a wide range of temporal and spatial scales. The multiple levels of organization within a complex system can mutually affect each other, giving rise to emergent global patterns (spatial and/ or temporal). Additionally, these systems are sensitive to different environmental cues, displaying remarkably specific responsiveness[1, 2]."

- Figure 1: DFA is super important to understand for the purpose of the paper, but it is not used commonly in biological rhythms research. I suspect the audience who'd be most interested in this work are chronobiologists and as such I think the authors could strengthen their manuscript by accommodating for the lack of familiarity. Figure 1 I and J are hard to interpret without further context. I don't think readers are going to have an intuitive sense of how to interpret these graphs in a meaningful way, especially because they are on different scales. Of course there is the supplementary information, but while it provides a lot of technical details of how DFA works, I don't think it or the results text really help the reader who is not a DFA expert or in the process of becoming one "get it"

Rsp: Thank you for your helpful comment. We have included a sentence that we think will be helpful to a diverse audience to familiarize with DFA, as follows (Line 151-153):

"DFA is a method for characterizing the scaling behavior (i.e. the type of autocorrelation properties) present in a time series (see details in SupplementarySection 4)."

Riggle et al. 2023 recently did somewhat similar work and used a strategy of creating artificial timeseries with known temporal components and performing their analyses on these as way to give their analyses context. I wonder if a similar approach would work well here. Creating artificial time series that look like real data, but have a a high level of autocorrelation or low I think could provide good contrast to understand the real data and should be minimal work to produce.

Rsp: We have made a new supplementary figure (Supplementary Figure 12) with artificial timeseries to help understand better long and short range correlations . The previous Supplementary Figure 12 and 13 were unified in the new Supplementary Figure 13 (see Supplementary Section 4).

Additionally I think the color choice of the event/inter event makes it hard to parse on these figures.

Rsp: We have changed the colors.

-While I applaud the authors for explaining this concept of autocorrelations (363-380). I think it would benefit from another pass. I spotted both a typo "alfa" instead of the greek letter and I think an example of behavioral memory and and further explanation could be warranted.

Rsp: We have corrected the typo. We also have included a paragraph with further explanation of what we mean with behavioral memory, as follows (lines 368-392):

"Behavioral memory refers to the probability that a behavioral action at a given time point (i.e. running or eating events) strongly depends on previous events of the same behavior. Behavioral dynamical processes, described through time series, can exhibit positive (auto) correlations, anti-correlations or no correlations between behavioral time points. These reflect different types of behavioral dependencies over time, and, therefore, behavioral memory. Positive correlations indicate that an event in the present makes it more likely that the same event will occur in the future. Anti-correlations indicate that an event in the present makes it more unlikely that the same event will occur in the future. Correlations can also be characterized by the duration of these dependencies. Long-range (auto) correlations indicate that these temporal dependencies persist over several orders of temporal magnitude in time series. Thus, they can be mathematically associated with a power-law function (i.e. linear regions in Figure 1 i and j, characterized by a slope, α), which presents poor decay over time and is thus associated with long-term memory. Short-range (auto) correlations, on the contrary, imply fast temporal decay of temporal dependencies; therefore, the process can be considered as a short-term and mathematically associated with the quick exponential decay of correlations over time [25,26]. In this context, the difference between the short-range correlations ($\alpha = 0.5$) found in feeding and long-range correlations ($0.5 > \alpha > 1$) seen in wheel running resides in how correlations persist. α -values above 1 and below 0.5 are indicative of persistent strong correlations or anti-correlations, respectively [18]. Since α -values represents the micro-structure of behavioral dynamics and are unaffected by differences in the mean level of activity [10] they are often more sensitive to stress, aging and illnesses than traditional summary statistic measures (i.e. counts, means, etc).

-There is a typo on line 478: should be dynamics instead of dynamical

Rsp: Thank you for spotting it, we have corrected the typo (line 488)

-It might be helpful to provide a short definition/example of anti-correlations

Rsp: We have included a brief definition of anti-correlations in lines 368-392 (see at the detailed response provided above, in the previous item).

-The latter half of discussion 503-638 is great and is a good model for the authors to use elsewhere in the paper.

Rsp: Thank you for the comment.